# Combined MEK and JAK/STAT3 pathway inhibition effectively decreases SHH medulloblastoma tumor progression

Jamie Zagozewski[1,14], Stephanie Borlase[1,14], Brent J. Guppy[1], Ludivine Coudière-Morrison[1], Ghazaleh M. Shahriary[1], Victor Gordon[1], Lisa Liang[1], Stephen Cheng[1], Christopher J. Porter [2], Rhonda Kelley[3], Cynthia Hawkins [4,5], Jennifer A. Chan [6,7], Yan Liang [8], Jingjing Gong[8], Carolina Nör[9], Olivier Saulnier [9], Robert J. Wechsler-Reya [10], Vijay Ramaswamy [4,9,11,12] & Tamra E. Werbowetski-Ogilvie [1,13✉]

Medulloblastoma (MB) is the most common primary malignant pediatric brain cancer. We recently identified novel roles for the MEK/MAPK pathway in regulating human Sonic Hedgehog (SHH) MB tumorigenesis. The MEK inhibitor, selumetinib, decreased SHH MB growth while extending survival in mouse models. However, the treated mice ultimately succumbed to disease progression. Here, we perform RNA sequencing on selumetinib-treated orthotopic xenografts to identify molecular pathways that compensate for MEK inhibition specifically in vivo. Notably, the JAK/STAT3 pathway exhibits increased activation in selumetinib-treated tumors. The combination of selumetinib and the JAK/STAT3 pathway inhibitor, pacritinib, further reduces growth in two xenograft models and also enhances survival. Multiplex spatial profiling of proteins in drug-treated xenografts reveals shifted molecular dependencies and compensatory changes following combination drug treatment. Our study warrants further investigation into MEK and JAK/STAT3 inhibition as a novel combinatory therapeutic strategy for SHH MB.

[1] Department of Biochemistry and Medical Genetics, Rady Faculty of Health Sciences, University of Manitoba, Winnipeg, MB, Canada. [2] Ottawa Bioinformatics Core Facility, Ottawa Hospital Research Institute, Ottawa, ON, Canada. [3] Central Animal Care Services, University of Manitoba, Winnipeg, MB, Canada. [4] The Arthur and Sonia Labatt Brain Tumour Research Centre, The Hospital for Sick Children, Toronto, ON, Canada. [5] Department of Laboratory Medicine and Pathobiology, University of Toronto, Toronto, ON, Canada. [6] Department of Pathology & Laboratory Medicine, University of Calgary, Calgary, AB, Canada. [7] Arnie Charbonneau Cancer Institute, University of Calgary, Calgary, AB, Canada. [8] Pathology Department, NanoString Inc, Seattle, WA, USA. [9] Developmental and Stem Cell Biology Program, The Hospital for Sick Children, Toronto, ON, Canada. [10] Sanford Burnham Prebys Medical Discovery Institute, La Jolla, CA, USA. [11] Division of Haematology/Oncology, The Hospital for Sick Children, Toronto, ON, Canada. [12] Department of Medical Biophysics, University of Toronto, Toronto, ON, Canada. [13] CancerCare Manitoba Research Institute, Winnipeg, MB, Canada. [14] These authors contributed equally: Jamie Zagozewski, Stephanie Borlase. ✉email: Tamra.Ogilvie@umanitoba.ca

Brain tumors are the deadliest form of childhood cancer and account for 20% of all new pediatric cancer cases. Medulloblastoma (MB) is the most common malignant primary pediatric brain tumor. Treatment currently consists of aggressive surgery, high doses of cytotoxic chemotherapy, and radiation to the whole brain and spinal cord. Unfortunately, 40% of patients still succumb to their disease, while survivors must deal with the long-term toxicities associated with chemotherapy and radiation[1–3].

MB consists of at least 4 highly distinct molecular subgroups: WNT, Sonic Hedgehog (SHH), Group 3, and Group 4[1,4]. SHH MBs are characterized by activation of the SHH pathway, and include very high-risk groups of both children (>3 years old) and infants (<3 years old) that exhibit significant intertumoral heterogeneity and account for the majority of treatment failures despite aggressive therapy[5–11]. For example, TP53 mutations confer a near universally fatal prognosis in older children with SHH MB tumors and constitute a very high-risk form of the disease[7]. Personalized therapies for SHH MB are lacking, as SHH pathway antagonists are not predicted to work in younger patients with tumors harboring mutations in downstream SHH pathway genes[12–15]. Moreover, premature osseous fusion[16] has limited the clinical utility of SHH pathway antagonists. Thus, novel targeted therapies are urgently needed.

A major contributing factor to MB cellular heterogeneity is the presence of putative brain tumor-propagating cells (TPC) or cancer stem/progenitor cells[17–20]. These primitive cells are believed to underlie progression and drug resistance in several cancers, including MB[19,21,22]. Therefore, the ability to identify robust TPC biomarkers and characterize their role in regulating MB tumorigenesis may reveal novel subgroup-specific therapies capable of targeting TPC subpopulations. We recently identified novel roles for the CD271/p75 neurotrophin receptor (CD271/p75NTR) and the MEK/ERK signaling pathway in contributing to SHH MB TPC growth and tumor progression[23,24]. Bioinformatics analyses of large patient sample datasets and tumorspheres from SHH MB cultures[24] demonstrated that CD271 is a potential diagnostic marker for these tumors. CD271+ cells exhibit upregulated MEK/ERK signaling and inhibiting this pathway reduced endogenous CD271 levels, stem cell proliferation, survival, and migration in vitro. In addition, CD271 knockdown decreased cell proliferation and pERK activity[23,24]. Importantly, treatment with the brain penetrant MEK inhibitor selumetinib extended survival and decreased CD271 levels in a biologically relevant preclinical animal model[24]. Combined with pERK staining in primary SHH MB tumor samples, our previous in vitro and in vivo findings demonstrated that the MEK/ERK pathway is a therapeutic target in human SHH MB. While these data were promising, selumetinib-treated mice ultimately succumbed to disease progression.

Thus, we sought to enhance the efficacy of selumetinib by targeting pathways that compensate for MEK inhibition in vivo with the overarching goal of further enhancing survival in SHH MB tumors. Here, using a combination of transcriptomics, proteomics, multiplex spatial profiling as well as in vitro and in vivo functional assays, we show that JAK/STAT3 activity is upregulated following selumetinib treatment in SHH MB tumor xenografts and that targeting this pathway, in combination with MEK inhibition, significantly attenuates tumor growth. Dual inhibition of MEK and JAK/STAT3 pathway activity abrogates tumorigenic properties in three different MB cell models in vitro as well as in 2 in vivo models of SHH MB. Thus, combinatory JAK/STAT3 and MEK pathway inhibition may serve as a novel therapeutic strategy for future SHH MB patients.

## Results

### JAK/STAT3 pathway activity is upregulated following MEK inhibition in vivo. Our previous work demonstrated that

selumetinib increased survival of NOD SCID mice bearing orthotopic SHH MB xenografts compared to vehicle controls[24]. However, the animals eventually succumbed to tumor progression[24]; thus, we sought to identify the altered molecular pathways that may compensate for MEK inhibition in vivo. We posit that these pathways can be therapeutically exploited to identify an optimized combinatorial therapeutic strategy to further decrease tumor growth.

To identify aberrant molecular changes following selumetinib treatment, we performed RNA sequencing (RNA-seq) on sorted control ($N = 3$ independent endpoint tumors at days 43 and 46 (2 mice) post-tumor cell injection) and selumetinib-treated ($N = 3$ independent endpoint tumors at days 56, 58 and 59 post-tumor cell injection) UI226 SHH MB cells (recently derived from a primary SHH MB tumor[23,25]) from tumor xenografts previously characterized[24] (Fig. 1a, b). Endpoint tumors from both the control and selumetinib groups were histologically similar displaying high nuclear-to-cytoplasmic ratios and well-encapsulated margins (Fig. 1b). We chose in vivo characterization over in vitro screen-based analyses, as this will identify pathways that may compensate for MEK inhibition in the more complex milieu of the cerebellar tumor microenvironment. RNA-seq revealed 576 significantly ($p < 0.05$) differentially expressed genes, the majority (79%) of which were upregulated (Fig. 1c, d, Supplementary Fig. 1a, and Supplementary Data 1) in selumetinib-treated cells. Genes associated with JAK/STAT3, TNFα /NFκB, as well as apoptotic signaling pathways were enriched in gene sets that were upregulated in selumetinib-treated xenografts (Fig. 1e and Supplementary Data 2–7). Importantly, genes that show decreased expression upon MEK activation were also enriched in upregulated gene sets in selumetinib-treated tumors, consistent with sustained decreased pathway activity. A total of 119 genes exhibited decreased expression in selumetinib-treated tumors (Fig. 1f). Of note, CD271 was the 3rd most significantly downregulated gene (5.2-fold) providing additional validation of our previous findings[24].

Very little is known about the role of the JAK/STAT3 pathway in human SHH MB tumor progression and drug resistance. Several drugs targeting JAK/STAT3 signaling have been identified and are known to be brain penetrant in xenograft models including pacritinib[26] and AZD1480[27]. Recent studies have also shown that STAT signaling activation accompanies MAPK pathway inhibition in melanoma[28,29], thyroid cancer[30], and breast cancer[31], thus implicating upregulated JAK/STAT3 activity as a general mechanism associated with drug resistance or long-term adaptive responses in multiple cancers. To further explore this pathway in our model system, we performed pSTAT3 (Tyr705) immunohistochemistry (IHC) on formalin-fixed paraffin-embedded (FFPE) tissue sections derived from control and selumetinib treated UI226 SHH MB xenografts. STAT3 is phosphorylated upon upstream JAK activation and is an appropriate readout of activated JAK/STAT3 signaling[32,33]. In line with the RNA-seq results, IHC revealed a significant 2.5-fold increase in the number of pSTAT3+ cells in selumetinib-treated xenografts relative to controls (Fig. 1g, h).

While we focused on in vivo molecular compensations following MEK inhibition, nevertheless, we examined pSTAT3 expression in SHH MB tumorspheres treated with selumetinib for 3 days in culture. pSTAT3 levels decrease with increasing selumetinib concentration, concomitant with an upregulation in apoptosis (Supplementary Fig. 1b). Given that our in vivo experiments were carried out long-term (30–40+ days treatment) in the more complex cerebellar tumor microenvironment, it is unsurprising that our short-term in vitro analysis did not recapitulate these findings. Therefore, we next cultured SHH MB tumorspheres over multiple passages using a lower, less toxic concentration of selumetinib to

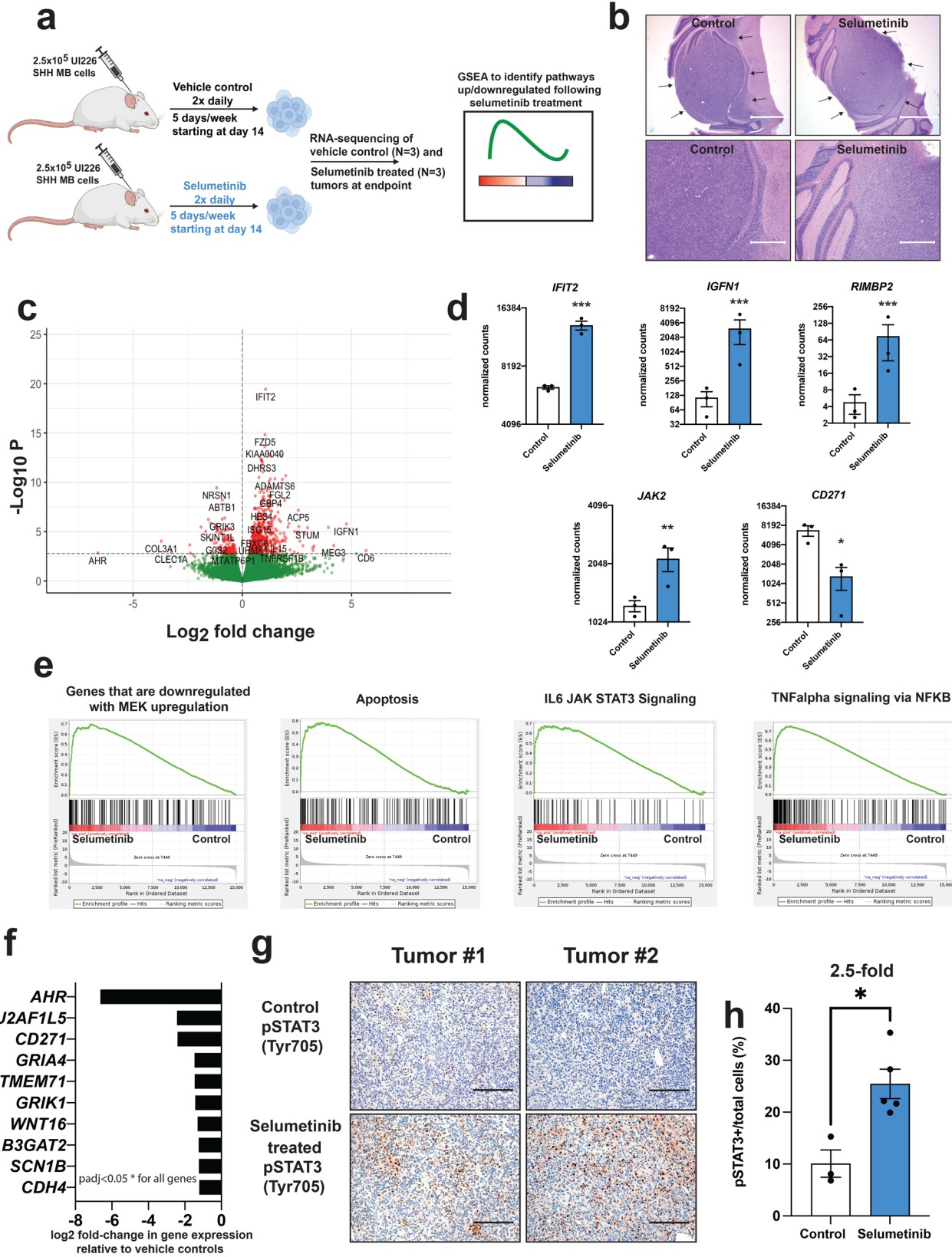

examine pSTAT3 levels following extended treatment. While Daoy tumorspheres are amenable to this long-term exposure and maintained over multiple passages, UI226 tumorspheres are not (Supplementary Fig. 1c). Indeed, under these conditions, Daoy tumorspheres also exhibit an increase in pSTAT3 levels following MEK inhibition (Supplementary Fig. 1d).

As lower pSTAT3 levels were observed in the UI226 xenograft vehicle controls in vivo, we next evaluated JAK/STAT pathway gene expression in primary MB patient samples. Analyses of Hallmark JAK/STAT signature genes[34] in a dataset consisting of 218 tumors across 7 types of pediatric brain cancers[35] demonstrate that the majority of JAK/STAT-related transcripts

**Fig. 1 Pathways and genes differentially expressed in selumetinib-treated xenografts. a** Schematic of workflow used to identify compensatory pathways in SHH xenograft tumors following MEK inhibitor treatment. A portion of this schematic, including mouse, syringe, and cell icon images, was created with BioRender.com. **b** Representative H&E images of FFPE sections from control (left) and selumetinib-treated (right) UI226 SHH MB xenografts. Scale bars: 1500 μm (upper) and 600 μm (lower). **c** Volcano plot depicting the log2 fold change and p-values in RNA-seq data with points in red identifying the 576 significantly differentially expressed genes (FDR < 0.05). **d** Normalized counts for representative differentially expressed transcripts from RNA-seq data. *IFIT2* ($p = 6.33E-16$), *IGFN1* ($p = 0.0002$), *RIMBP2* ($p = 0.00049$) and *JAK2* ($p = 0.0041$) are increased while *CD271* ($p = 0.047$) expression is decreased in selumetinib-treated UI226 SHH MB xenografts relative to vehicle controls. $p < 0.05$ *, $p < 0.01$**, $p < 0.001$***. Expression differences were calculated using the DESeq results function. Differentially expressed genes were identified using a q-value (Benjamini-Hochberg corrected p-value) cut-off of 0.05, which coincides with a p-value of 0.00168. **e** GSEA demonstrating that genes associated with a downregulated MEK signature, increased JAK/STAT3, TNFalpha, and apoptosis signaling are enriched in genes sets that are upregulated in selumetinib-treated xenografts. padj < 0.05* for all signatures. **f** Most significantly downregulated genes in selumetinib-treated xenografts. Differentially expressed genes were identified using a q-value (Benjamini-Hochberg corrected p-value) cut-off of 0.05, which coincides with a p-value of 0.00168. padj < 0.05* for all genes. **g** Representative images of IHC staining for pSTAT3 (Tyr705) in FFPE sections from two independent control (upper) and selumetinib-treated (lower) UI226 SHH MB xenografts. Scale bar: 150 μm. **h** Quantitative analyses of IHC pSTAT3 (Tyr705) staining in vehicle control (white) and selumetinib-treated (blue) xenografts. The proportion of pSTAT3+ cells for each sample was calculated using QuPath and expressed as the total number of pSTAT3+ cells relative to the total number of cells in each image (pSTAT3- nuclei (hematoxylin-stained) and pSTAT3+ cells). Significance was calculated using a two-tailed t-test. Error bars: SEM. $p = 0.011$.

and proteins are lower across the 22 MB samples (Fig. 2a). However, the known negative regulator of JAK/STAT signaling, protein tyrosine phosphatase non-receptor type 2 (PTPN2), is strikingly higher across the MB subset (Fig. 2a). More specific hierarchical clustering selecting genes restricted to JAK/STAT pathway gene sets across an independent cohort of 763 MB patient samples stratified MB tumors into the SHH subgroup (Fig. 2b and Supplementary Fig. 2). These transcriptome signatures are significantly lower in SHH MBs relative to the other subgroups (Fig. 2c). Collectively, these data reveal low basal JAK/STAT pathway activity in both vehicle control xenografts and primary patient MB samples with the lowest levels in SHH MB tumors. Therefore, JAK/STAT3 signaling activation is a selumetinib-induced response in our UI226 SHH MB model.

**Treatment with JAK/STAT3 pathway inhibitors significantly inhibits tumor properties in vitro.** To initially test the effect of inhibiting the JAK/STAT3 pathway on tumor properties, we developed a high-content 3D tumorsphere drug-screening assay to construct high-resolution concentration-response curves following treatment with the individual JAK/STAT3 inhibitors pacritinib and AZD1480 in vitro (Fig. 3a). Serum-free tumorsphere assays were specifically employed, as pSTAT3 is expressed in these stem cell-enriched, defined culture conditions. pSTAT3 is markedly decreased following treatment with both JAK/STAT3 inhibitors (Fig. 3b–d and Supplementary Fig. 3a–c). Similar decreases in pSTAT3 levels were observed in tumorspheres from two different human cell lines, UI226 and Daoy[36], as well as *Ptch+/−:p53+/−* SHH mouse MB cultures derived from primary tumors. These mouse SHH MB cells are amenable to tumorsphere culture and retain SHH signaling in either stem-cell enriched or modified stem-cell conditions[37,38]. We therefore utilized the tumorsphere assay to systematically score thousands of tumorspheres at various drug concentrations over 5 days using sphere size as a readout. To avoid inclusion of dead cells and aggregates, only true tumorspheres with diameters between 45 and 360 μm and circularities greater than 0.3 were scored. The results (Fig. 3e–f) were then used to determine the optimal μM range of concentrations for use in additional cellular assays. For comparison, we also evaluated the chemotherapeutic agent vincristine, which is quite potent in the nM range, in our tumorsphere model (Supplementary Fig. 3d).

The inhibitory effects of pacritinib and AZD1480 on tumorsphere size were independently validated by cumulative frequency distribution analyses using Fiji/ImageJ for both UI226 and Daoy (Fig. 3g, h and Supplementary Fig. 3e, f). At the highest doses, we observed reductions in tumorsphere number for all 3 cell models when treating with pacritinib and/or AZD1480 (Supplementary

Fig. 3g–l). However, at these concentrations, there were no statistically significant reductions in cell viability (Supplementary Fig. 3m–r). When all 3 cell models were cultured as aggregates in 3D collagen to assess migration, significant decreases in cell motility were observed following treatment with both drugs (Fig. 3i–n). Collectively, these results show an overall decrease in tumorigenic properties of MB cells including migration, tumorsphere size, and number following treatment with single JAK/STAT3 inhibitors in vitro.

**Dual MEK and JAK/STAT3 pathway inhibition elicits a further reduction in SHH MB tumorigenic properties in vitro.** Next, we tested the hypothesis that simultaneous JAK/STAT3 + MEK inhibition would yield a further reduction in tumorsphere size in our high-content 3D tumorsphere assay. To evaluate in vitro drug synergy, we generated drug combination matrices in 96-well plates such that each well contained a unique combination of AZD1480 + selumetinib or pacritinib + selumetinib. Both JAK/STAT3 inhibitors produced a synergistic reduction in tumorsphere size when combined with selumetinib. Specifically, a concentration of 2.5 μM selumetinib in combination with 39 nM or 156 nM pacritinib elicited a synergistic reduction in both UI226 and Daoy tumorsphere size (Fig. 4a, b). When combining selumetinib with AZD1480, drug synergies were also observed at several concentrations (Fig. 4a, b). Additional reductions in tumorsphere size were independently validated by cumulative frequency distribution analyses using Fiji/ImageJ for both UI226 and Daoy cells (Fig. 4c, d).

Having established a decrease in tumorsphere size following MEK + JAK/STAT3 inhibitor treatment, we sought to determine if loss of cell viability, as assessed by Trypan blue exclusion assays, could account for the smaller MB tumorspheres (Supplementary Fig. 4a–c). A significant reduction in cell viability was observed following combination pacritinib + selumetinib treatment for both UI226 and Daoy tumorspheres (Supplementary Fig. 4a, b). AZD1480 + selumetinib also significantly reduced cell viability for UI226 but not for Daoy tumorspheres (Supplementary Fig. 4a, b). Similarly, Annexin V staining of both pacritinib + selumetinib and AZD1480 + selumetinib treated UI226 tumorspheres revealed significant increases in the proportion of dead cells (Annexin V+/7AAD+) compared to single-agent treatment (Supplementary Fig. 4d, e). However, there was no change in *Ptch+/−:p53+/−* SHH MB cell viability following combination drug treatment (Supplementary Fig. 4c) suggesting that MEK + JAK/STAT3 inhibition results in either cytostatic or a combination of cytostatic/cytotoxic effects on MB tumorspheres.

Finally, we evaluated the effect of dual MEK + JAK/STAT3 inhibitor treatment on migration. Statistically significant further

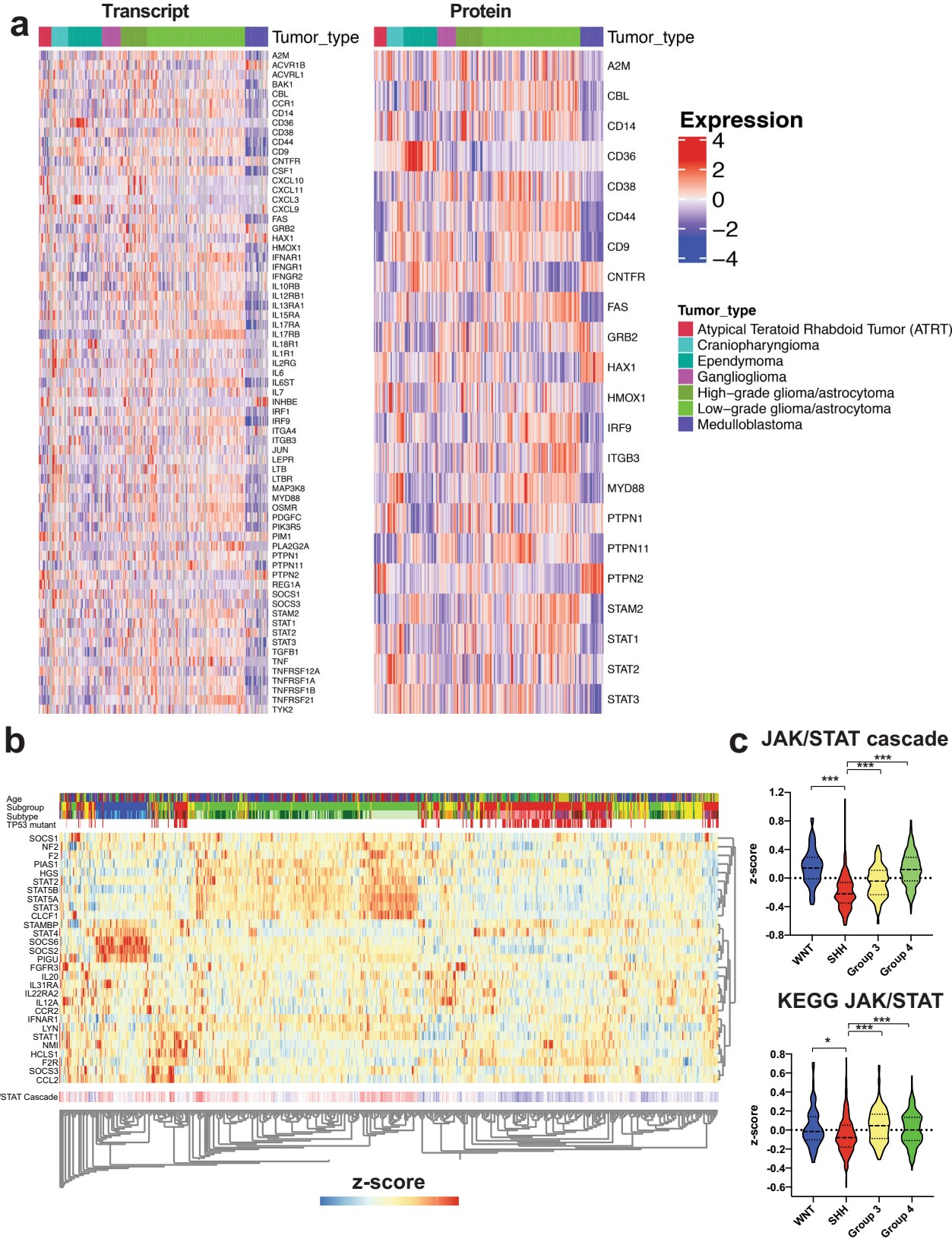

reductions in migration compared to single agents and/or DMSO controls were observed for UI226, Daoy and *Ptch*+/−:*p53*+/− MB cells (Fig. 4e–h). Taken together, our in vitro assays demonstrate that dual MEK + JAK/STAT3 inhibition results in further reductions in SHH MB cell tumorigenic properties in vitro.

**RNA sequencing reveals significant molecular changes in SHH MB tumorspheres following MEK inhibition or dual MEK and JAK/STAT3 pathway inhibition.** As dual JAK/STAT3 + MEK inhibition reduced tumorsphere size and cell migration, we next performed RNA-seq on selumetinib and/or pacritinib treated UI226 tumorspheres to characterize the sustained transcriptomic

**Fig. 2 The JAK/STAT3 signaling signature is significantly lower in SHH MB primary patient tumors. a** Heatmap generated by ProTrack [http://pbt.cptac-data-view.org/] illustrating transcript (left) and protein (right) levels of Hallmark IL6 JAK/STAT3 signaling signature components across 218 samples representing several classes of childhood brain cancer. Individual samples are aligned vertically into columns, with the 22 MB samples colored purple. JAK/STAT3 signaling components were retrieved from Supplementary Data 5. Signaling components absent from the ProTrack database are omitted from visualization. **b** Unsupervised hierarchical clustering restricted to the JAK/STAT signaling cascade-associated transcripts across the 4 MB subgroups from the Cavalli et al.[5] dataset representing 763 patient samples (WNT: $N = 70$ samples in blue; SHH: $N = 223$ samples in red; Group 3: $N = 144$ samples in yellow; Group 4: $N = 326$ samples in green). **c** Violin plots depicting z-scores for 2 JAK/STAT signaling cascade transcriptome signatures across the 4 MB subgroups. Significance was determined by ANOVA with a Tukey's test for multiple comparisons. $p < 0.05$*, $p < 0.001$***. For all violin plots, the center line is the median, the 25th and 75th percentiles are depicted by the lower and upper dotted lines, respectively, and the extremes of the distribution extend to the minima and maxima.

changes 5 days following treatment in vitro. Interestingly, the pacritinib-treated tumorspheres at day 5 resembled the vehicle controls with only 9 sustained significantly differentially expressed genes (Fig. 5a). This suggests that the molecular changes associated with pacritinib alone are short-lived. In contrast, one-time selumetinib treatment was sufficient to drive the major transcriptomic changes at day 5 with 6628 genes exhibiting significant differential expression (FDR < 0.05) (Fig. 5b). In line with these findings, both selumetinib and combination drug treatment resulted in similar changes to hallmark gene sets, canonical pathways and oncogenic signatures (Fig. 5b–g). Genes/pathways associated with cell cycle and DNA replication/repair were the most significantly enriched downregulated gene sets following selumetinib or combination drug treatment (Fig. 5d). Strikingly, these pathways have also been shown to be strongly enriched specifically in high-risk SHH MB patient tumors exhibiting *TP53* mutations[5,7]. In addition, genes/pathways associated with neurogenesis and epithelial to mesenchymal transition (EMT) were significantly enriched among genes that were downregulated following selumetinib or combination drug treatment (Fig. 5d, e).

Further exploring the transcriptomic changes related to cell fate, we observed that gene sets associated with granule neuron progenitor cell proliferation after SHH stimulation were enriched for genes that were downregulated following selumetinib or combination drug treatment (Fig. 5f, upper). While most genes were not specifically associated with SHH signaling, significant changes in *PTCH1*, *SUFU* and *GLI3* were observed (Fig. 5f, lower). In contrast, gene sets associated with NOTCH signaling, astrocytes, and neural crest differentiation (Fig. 5d, g) were significantly enriched among genes that were upregulated following combination drug treatment and/or selumetinib alone, likely reflecting the sustained, but indirect, effects after 5 days. Taken together, these results demonstrate that the transcriptome changes at day 5, in vitro, are mainly driven by the sustained effects of selumetinib treatment.

**Dual MEK + JAK/STAT3 pathway inhibition significantly improves survival and reduces tumor growth in vivo.** Next, we examined whether combinatorial drug treatment could result in a further reduction of tumor growth in vivo. As pacritinib has been shown to be effective at much lower concentrations, we focused on this drug in combination with selumetinib. Initial pilot studies with UI226 were completed to determine the maximum tolerated dose and revealed that 25, 50, and 100 mg/kg of pacritinib were all well-tolerated with no toxicities when administered via oral gavage twice daily, 5 days a week (Fig. 6a). Pacritinib treatment alone did not extend survival in vivo (Fig. 6b). Given the overall lower levels of pSTAT3 specifically in the UI226 vehicle controls (Fig. 1g, h), this was not surprising. Based on these findings, 50 mg/kg pacritinib was chosen for combination therapy in vivo along with a reduced dose of 37.5 mg/kg selumetinib (compared to 75 mg/kg selumetinib in Fig. 1 and our previous work[24]).

Following tumor engraftment, mice were administered vehicle, 50 mg/kg pacritinib, 37.5 mg/kg selumetinib, or a combination of 50 mg/kg pacritinib and 37.5 mg/kg selumetinib twice daily, 5 days a week until endpoint was reached. This treatment schedule resulted in highly favorable toxicity profiles, as weights steadily increased, even with pacritinib + selumetinib treatment (Fig. 6c). Importantly, the combination of pacritinib + selumetinib significantly improved survival in our UI226 xenograft model relative to vehicle control and each drug alone (Fig. 6d). In contrast, no significant improvements in survival were observed when HDMB03 Group 3 MB xenografts were treated with selumetinib and/or pacritinib relative to the vehicle (Fig. 6e). To compare differences in tumor growth across treatment groups, a subset of UI226 vehicle and treated xenografts was also extracted prior to endpoint, all at day 41, and samples were stained for the human specific marker STEM121 (Fig. 7a, b). While a small decrease in tumor size was observed for single agent therapies, combination drug treatment resulted in a substantial and significant reduction in tumor growth (Fig. 7a–c). Similarly, combination drug treatment of RCMB18 SHH MB patient-derived xenografts (PDX) in a small cohort of NOD SCID IL2Rg null (NSG) mice significantly reduced tumor growth, as single well-encapsulated nodules were observed relative to vehicle controls (Fig. 7d, e). Interestingly, this growth pattern was observed in the presence of pacritinib, with or without selumetinib. The tumors typically appeared as single well-encapsulated nodules as opposed to the vehicles and selumetinib-only samples which consisted of multiple infiltrating masses (Fig. 7d). Collectively, these data demonstrate that pacritinib + selumetinib treatment attenuates tumor progression in two biologically relevant SHH MB in vivo models.

**Digital spatial profiling reveals unique protein changes following in vivo treatment.** To more comprehensively and quantitatively explore the molecular changes following combination drug treatment, we chose to employ multiplex digital spatial profiling, an approach that combines protein abundance and spatial distribution data using antibodies coupled to photo-cleavable oligonucleotide tags[39]. Once unstained FFPE slides are labeled, the oligonucleotides are released from select regions of interest (ROIs) by UV exposure and quantified[39]. Identification of protein/phospho-protein differences between samples provided immediate insight into the long-term effects of our inhibitors. Using this method, we profiled 56 proteins, including a neural cells core panel, a MAPK, a PI3K as well as a cell death protein panel, across 12 ROIs for representative vehicle, pacritinib, selumetinib, and combination therapy (pacritinib + selumetinib) UI226 FFPE sections collected prior to endpoint. For each sample, 6 ROIs in the tumor core and 6 ROIs along the tumor/mouse brain boundary were interrogated (Fig. 8a–d). CD271, MAP2, and Ki67 were used to visualize the tumor regions by immunofluorescence (Fig. 8a–d). Quantification of CD271 levels across all 12 ROIs/sample showed no significant changes following

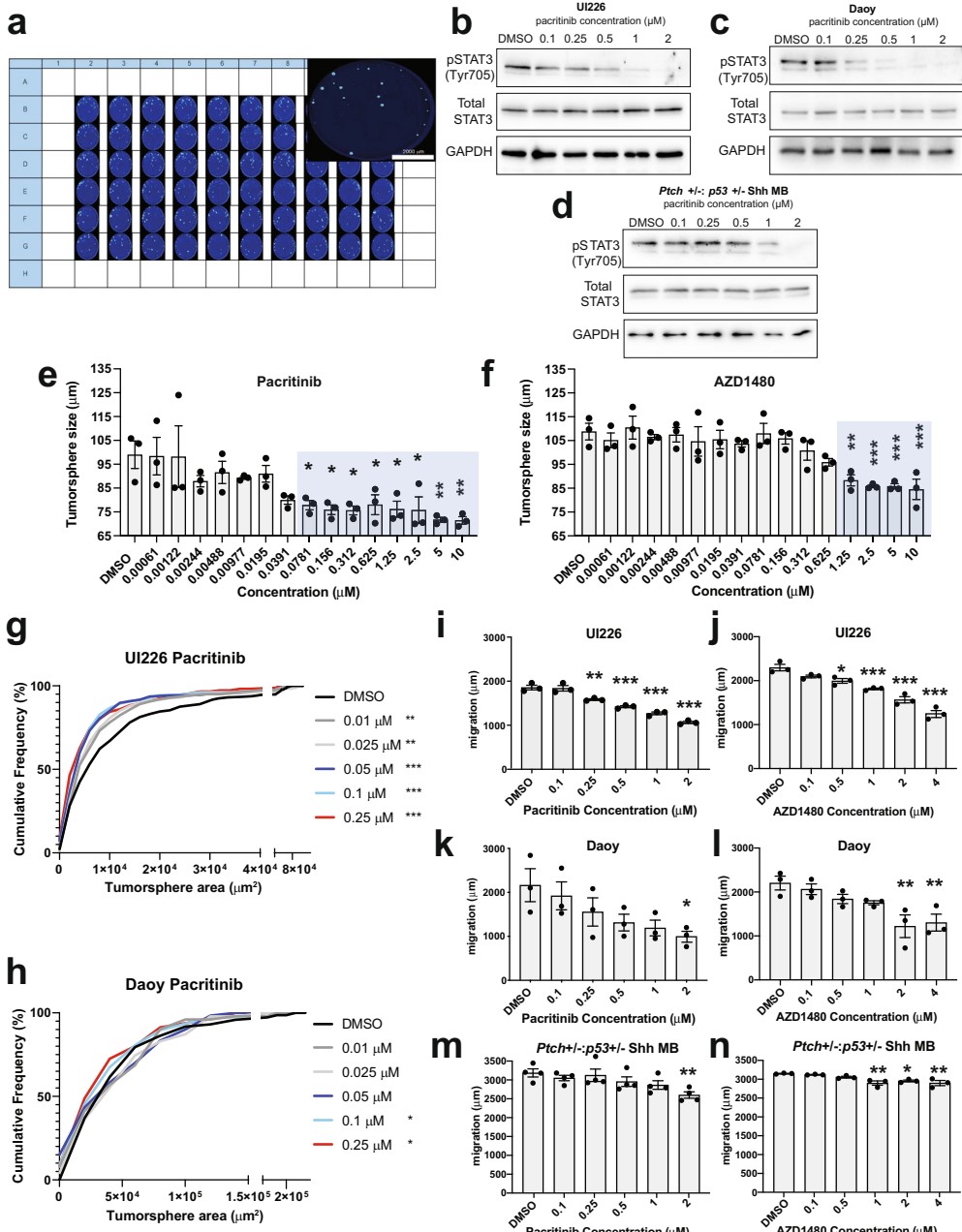

**Fig. 3 JAK/STAT3 inhibitors decrease tumorsphere size and cell migration in vitro. a** High-content tumorsphere drug screening set-up and 96-well plate depicting Hoechst 33342 stained tumorspheres following treatment with various drug concentrations. Approximately 4000 UI226 cells were seeded into each well of a 96-well ultra-low attachment plate. Cells were seeded in various drug concentrations or in the presence of vehicle (DMSO at 0.0025%) at 37 °C for 5 days. Following incubation, tumorspheres were fixed and stained with a 4% formaldehyde + 300 ng/mL Hoechst 33342 solution at 4 °C O/N. Western blot for pSTAT3 (Tyr705) activation, total STAT3 and GAPDH following treatment of UI226 (**b**) Daoy (**c**) or *Ptch+/−:p53+/−* (**d**) MB tumorspheres with pacritinib for 3 h. UI226 tumorsphere size following treatment with increasing doses of pacritinib (**e**) and AZD1480 (**f**). Error bars: SEM. $N = 3$ biological replicates and $n = 3$ technical replicates for each biological replicate. Results were analyzed using one-way ANOVA and a Dunnett's test for multiple comparisons. $p < 0.05^*$, $p < 0.01^{**}$, $p < 0.001^{***}$. For pacritinib treated cells in (**e**): 0.0781 μM, $p = 0.0432$; 0.156 μM, $p = 0.0221$; 0.312 μM, $p = 0.0200$; 0.625 μM, $p = 0.0459$; 1.25 μM, $p = 0.0244$; 2.5 μM, 0.0211; 5 μM, $p = 0.0046$; 10 μM, $p = 0.0042$. For AZD140 treated cells in (**f**): 1.25 μM, $p = 0.0016$; 2.5 μM, $p = 0.0004$; 5 μM, $p = 0.0004$; 10 μM, $p = 0.0002$. Cumulative frequency distribution of tumorsphere size for UI226 (**g**) and Daoy (**h**) following treatment with various concentrations of pacritinib. Tumorsphere size was analyzed using 2-sample Kolmogorov–Smirnov tests. $p < 0.05^*$, $p < 0.01^{**}$, $p < 0.001^{***}$. For pacritinib treated UI226 tumorspheres in (**g**): 0.01 μM, $p = 0.0073$; 0.025 μM, $p = 0.0017$; 0.05 μM, 0.1 μM and 0.25 μM, $p < 0.0001$. For pacritinib treated Daoy tumorspheres in (**h**): 0.1 μM, $p = 0.0277$; 0.25 μM, $p = 0.0107$. UI226 (**i, j**), Daoy (**k, l**) and *Ptch+/−:p53+/−* MB (**m, n**) cells exhibit a decrease in cell migration following treatment with pacritinib (**i, k, m**) and AZD1480 (**j, l, n**). Error bars: SEM. $N = 3$ or 4 biological replicates. Results were analyzed using one-way ANOVA and a Dunnett's test for multiple comparisons. For UI226/pacritinib treated cells (**i**): 0.25 μM, $p = 0.0041$; 0.50, 1.0 and 2.0 μM pacritinib, $p < 0.0001$. For UI226/AZD1480 treated cells (**j**): 0.5 μM, $p = 0.0139$; 1.0 μM, $p = 0.0004$; 2.0 and 4.0 μM, $p < 0.0001$. For Daoy/pacritinib treated cells (**k**): 2 μM, $p = 0.0362$. For Daoy/AZD1480 treated cells (**l**): 2.0 μM, $p = 0.0041$; 4.0 μM, $p = 0.0078$. For Mouse SHH MB/pacritinib treated cells (**m**): 2.0 μM, $p = 0.0084$. For Mouse SHH MB/AZD1480 treated cells (**n**): 1.0 μM, $p = 0.0028$; 2.0 μM, $p = 0.0167$; 4.0 μM, $p = 0.0029$.

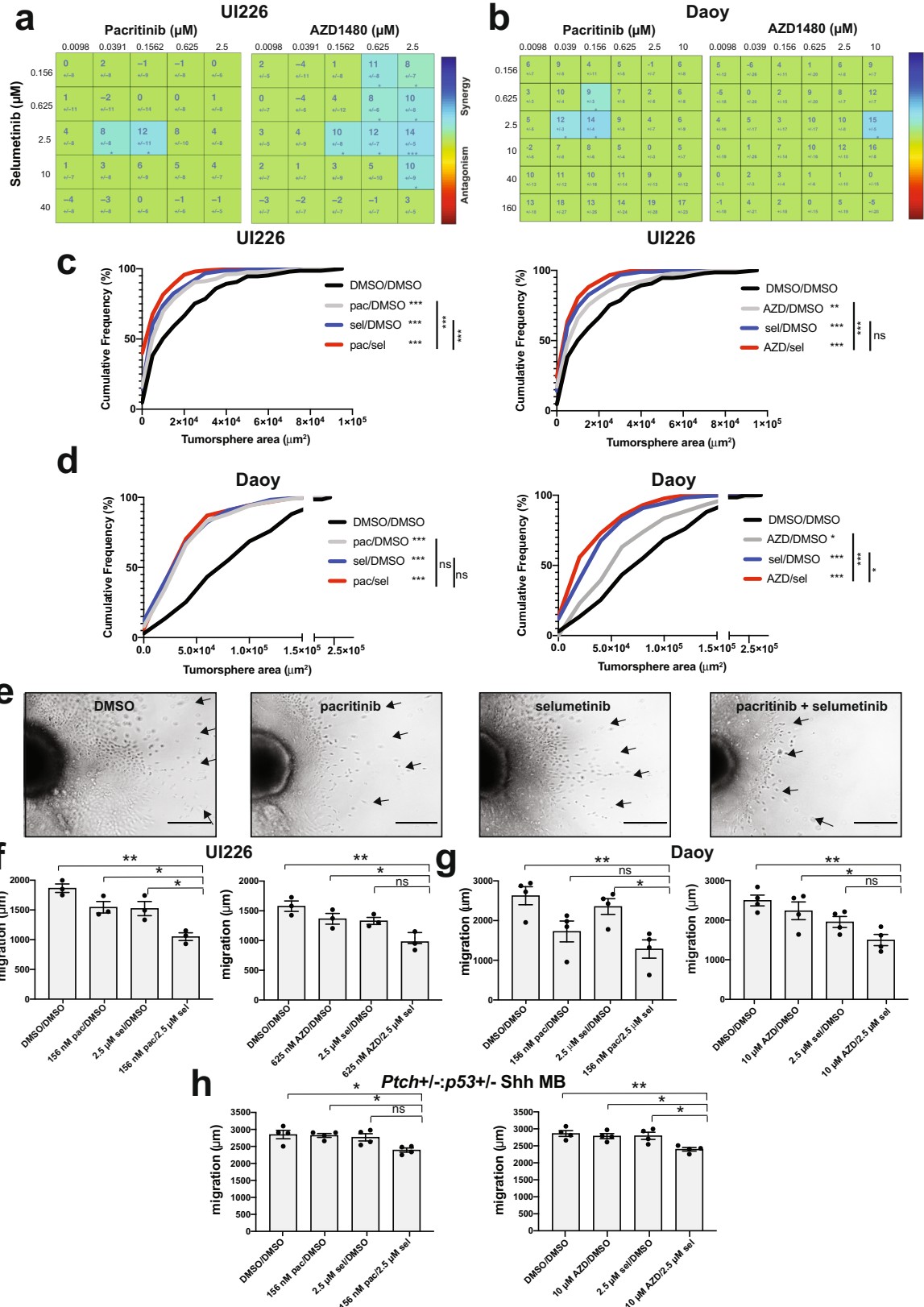

pacritinib treatment (Fig. 8e, f). However, a significant down-regulation was observed after selumetinib treatment and a further reduction was seen with combination therapy (Fig. 8e, f). These results support our current (Fig. 1) and previous findings demonstrating decreased CD271 levels following selumetinib treatment[24] across platforms.

While our tumor samples were positive for the human cell marker STEM121 (Fig. 7a), we cannot exclude the possibility that there are mouse cells infiltrating the tumor, particularly along the border. As a significant portion of the 56 antibodies cross-react with mouse antigens, we next compared the 56 chosen proteins in the tumor core vs. the border across all 4 samples. No significant

**Fig. 4 Combination of MEK + JAK/STAT3 inhibition significantly inhibits SHH MB tumorsphere size, and cell migration. a, b** Combenefit synergy plots for UI226 (a) and Daoy (b) tumorspheres for pacritinib + selumetinib and AZD1480 + selumetinib. Synergy plots represent $N = 3$ biological replicates and $n = 9$ technical replicates. Numbers in the synergy matrices represent the percentage in tumorsphere size reduction that is greater than predicted by an additive model ± SD. Drug combinations are color-coded as Blue: Synergy, Green: Additive, and Red: Antagonism. Cumulative frequency distribution of tumorsphere size for UI226 (**c**) and Daoy (**d**) following treatment with pacritinib + selumetinib or AZD1480 + selumetinib. Tumorsphere size was analyzed using two-sample Kolmogorov–Smirnov tests. $p < 0.05$*, $p < 0.01$**, $p < 0.001$***. For UI226 pacritinib combination in (**c**): all comparisons, $p < 0.0001$. For UI226 AZD1480 combination in (**c**): DMSO vs AZD, $p = 0.0078$; DMSO vs sel and DMSO vs AZD/sel, $p < 0.0001$; AZD vs AZD/sel, $p = 0.0004$. For Daoy pacritinib combination in (**d**): DMSO vs all treatment groups, $p < 0.0001$. For Daoy AZD1480 combination in (**d**): DMSO vs AZD, $p = 0.0125$; DMSO vs sel and DMSO vs. AZD/sel, $p < 0.0001$; AZD vs AZD/sel, $p < 0.0001$; sel vs AZD/sel, $p = 0.031$. **e** Representative images of migration through collagen type I gels following pacritinib, selumetinib or pacritinib + selumetinib treatment over 3 days. Scale bar: 400 µm. Quantification of cell migration in UI226 (**f**), Daoy (**g**) and *Ptch+/−:p53+/−* SHH MB (**h**) aggregates to assess the effects of pacritinib + selumetinib or AZD1480 + selumetinib treatment. Error bars: SEM. $N = 3$–4 independent biological replicates for each. Results were analyzed using ANOVA followed by a Tukey's test for multiple comparisons. **f** For UI226 pacritinib combination: DMSO vs pac/sel, $p = 0.0010$; pac vs pac/sel, $p = 0.0202$; sel vs pac/sel, $p = 0.0255$. For UI226 AZD1480 combination: DMSO vs AZD/sel, $p = 0.0043$; AZD vs AZD/sel $p = 0.0457$. **g** For Daoy pacritinib combination: DMSO vs pac/sel, $p = 0.0069$; sel vs pac/sel, $p = 0.0295$. For Daoy AZD1480 combination: DMSO vs AZD/sel, $p = 0.0049$; AZD vs AZD/sel, $p = 0.0345$. **h** For mouse SHH MB pacritinib combination: DMSO vs pac/sel, $p = 0.0198$; pac vs pac/sel, $p = 0.0280$. For mouse SHH MB AZD1480 combination: DMSO vs AZD/sel, $p = 0.0079$; AZD vs AZD/sel, $p = 0.0254$; sel vs AZD/sel, $p = 0.0220$.

differences were observed for the vehicle, pacritinib, and combination-treated samples, while only 3 proteins showed a significant difference in the selumetinib sample (Supplementary Fig. 5a–d). This was in stark contrast to our recently published studies for Group 3 MB in which mTOR inhibitor treatment induced neural differentiation regionally along the tumor border (Supplementary Fig. 5e–h)[40]. As we did not observe these regional differences in the current study, all 12 ROIs/sample were combined for additional analyses.

Multiplex spatial profiling revealed significant differences between the vehicle and inhibitor-treated samples (Fig. 8g–s and Supplementary Data 8–10). As expected, the selumetinib-treated tumor exhibited significant changes in proteins associated with the MAPK pathway (Fig. 8h). This includes downregulated pMEK1, total ERK1/2, and BRAF (Fig. 8j–k, m) but not pERK1/2 (Fig. 8l). There were no consistent and significant increases in differentiation proteins following treatment (Fig. 8g–i, p, q), but a reduction in OLIG2 was observed following selumetinib treatment alone (Fig. 8p). OLIG2 + progenitor cells have recently been shown to represent a putative tumor-initiating cell population in mouse models of SHH MB and are also enriched in human SHH MB stem-like cells[41]. Interestingly, sustained decreases in pMEK, total ERK, and BRAF were not observed following combination drug treatment; however, there was a significant reduction in pERK as well as significant upregulation of neurofibromin 1 (NF1), a tumor suppressor that reduces Ras activity and thus MAPK signaling (Fig. 8j–n). Of note, p53 was most significantly upregulated following combination treatment (Fig. 8o). In the pacritinib-treated tumor, no changes in pMEK, total ERK, and BRAF were observed; however, pERK was downregulated and NF1 was significantly upregulated (Fig. 8j–n), likely reflecting pathway crosstalk. Importantly, across all treatment groups, there were significant decreases in proteins associated with PI3K/AKT signaling such as PLCG1 and pAKT1 (Fig. 8r–s), again suggesting suppression through pathway crosstalk. As CD271 levels were sustained across several ROIs following pacritinib treatment (Fig. 8e, f), the data for this sample were also re-analyzed to compare protein levels between CD271^high and CD271^low ROIs; however, there were no significant differences between these groups for the proteins evaluated (Supplementary Fig. 6). Collectively, these results demonstrate the value of multiplex protein profiling in rapidly characterizing drug responses in xenograft models of SHH MB. While all 3 drug-treated samples exhibited significant alterations in MAPK and PI3K/AKT-associated proteins, differences at the individual protein level between samples and changes in molecular dependencies following treatment reveal compensatory changes in residual cells to sustain tumor growth suppression.

## Discussion

Current large-scale sequencing efforts rely on sampling from bulk SHH MB tumors. While these technologies have revolutionized our understanding of intertumoral heterogeneity between the subgroups and have improved patient stratification, they do not take into account the extensive intratumoral heterogeneity, namely the putative stem cells, progenitors, and more differentiated phenotypes observed in these tumors. By deconstructing cellular heterogeneity and the mechanisms regulating self-renewal and differentiation, we recently revealed novel molecular vulnerabilities in specific subpopulations (ie. CD271+ cells) that could be exploited using brain penetrant small molecule inhibitors[24]. Combined with pERK staining in primary patient samples, this led to the demonstration that the MEK/ERK pathway is a novel therapeutic target in human SHH MB models. Indeed, previous studies have shown that MEK/ERK signaling drives SHH pathway inhibitor resistance and promotes metastasis specifically in SHH MB mouse models[38]. However, single-agent therapies typically fail due to the upregulation of compensatory pathways. Thus, we performed RNA-seq on selumetinib-treated xenograft tumors in lieu of in vitro drug screens to identify adaptive responses in a more biologically relevant model system. We discovered that JAK/STAT3 pathway activity is elevated following single agent selumetinib therapy leading to a comprehensive examination of combinatorial MEK + JAK/STAT3 inhibitor treatment. Our in vitro findings demonstrate significant reductions in tumorigenic properties of SHH MB, including tumorsphere size and migration, with combination MEK + JAK/STAT3 inhibitor treatment compared to controls and monotherapies. Importantly, pacritinib + selumetinib treatment resulted in a significant reduction in tumor burden in 2 in vivo models suggesting that combination treatment could potentially be used as a tumor debulking strategy for SHH MB patients.

Recently, pacritinib has been examined for its efficacy in the treatment of glioblastoma (GBM)[26]. While the reduction in tumorigenic properties was observed in vitro, pacritinib alone was insufficient to increase the survival of orthotopic xenograft models in vivo, further underscoring the need for combination-based therapeutic approaches. Indeed, combination treatments with pacritinib and the tyrosine kinase inhibitor, afatinib, have also recently been evaluated in GBM[42]. Toxicity issues inhibited evaluation of survival in the combination treatment group in vivo; however, a significant reduction in tumor burden was observed following pacritinib + afatinib treatment compared to single-agent therapies and controls[42]. Of note, for the combination therapy studies, we chose 50 mg/kg pacritinib as opposed to the 100 mg/kg utilized in previous brain tumor studies[26,42].

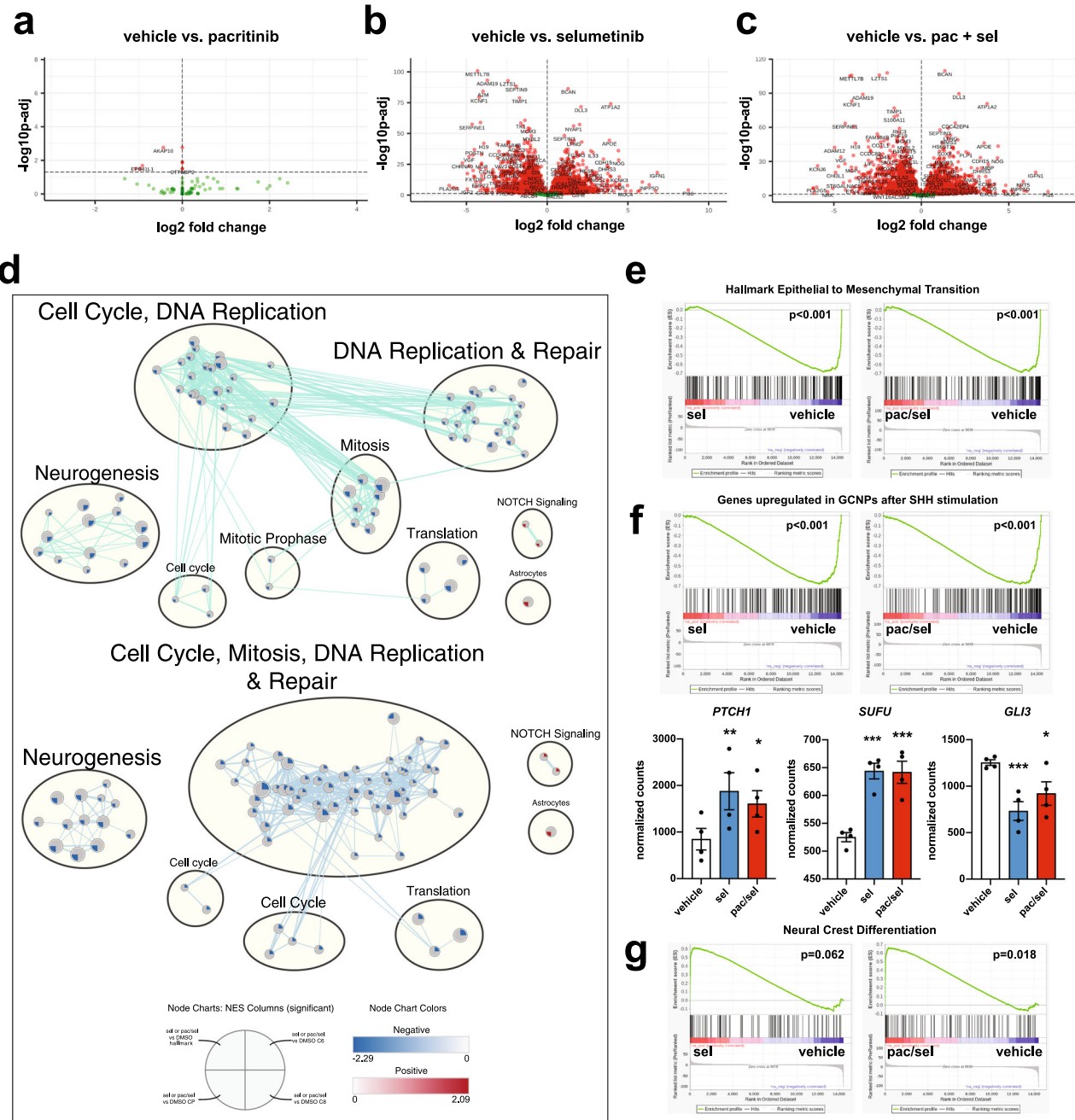

**Fig. 5 MEK or MEK + JAK/STAT3 inhibition significantly alters the transcriptome of UI226 SHH MB-treated tumorspheres.** Volcano plot depicting the log2 fold change and p-values in RNA-seq data comparing pacritinib (N = 6 samples) (**a**), selumetinib (N = 4 samples) (**b**) or pacritinib + selumetinib (N = 4 samples) (**c**) to vehicle controls (N = 4 samples) (FDR < 0.05). **d** Enriched pathways and biological processes as determined by GSEA following selumetinib (upper) or pacritinib + selumetinib (lower) treatment. GSEA results for Hallmark, Canonical Pathways (C2:CP), Oncogenic Signatures (C6), and Cell Type Signatures (C8) (q-value <0.005) were used to build the enrichment map. Node coloring is based on the normalized enrichment score (NES), with blue representing enrichment in vehicle and red representing enrichment in selumetinib or pacritinib + selumetinib (NOTCH signaling and Astrocyte signature only). **e** Genes associated with epithelial to mesenchymal transition are enriched in gene sets that are downregulated in the selumetinib and pacritinib + selumetinib treated tumorsphere samples relative to vehicle controls. p < 0.001***. **f** GSEA (upper) and normalized counts (lower) depicting changes in genes associated with granule neural precursors cells following SHH stimulation (upper) or in specific SHH signaling pathway genes (lower) following selumetinib or pacritinib + selumetinib treatment. For differentially expressed genes (lower), multiple testing correction was performed using the Benjamini Hochberg method, and genes identified using a q-value (i.e. a corrected p value) cut-off of 0.05. p < 0.05*, p < 0.01**, p < 0.001***. For *PTCH1*: sel, p = 0.0082; pac/sel, p = 0.0235. For *SUFU*: sel, p = 2.21E-05; pac/sel, p = 2.92E-05. For *GLI3*: sel, p = 0.00079; pac/sel, p = 0.04567. **g** Genes associated with neural crest differentiation are enriched in gene sets that are upregulated in the pacritinib + selumetinib treated samples relative to vehicle controls.

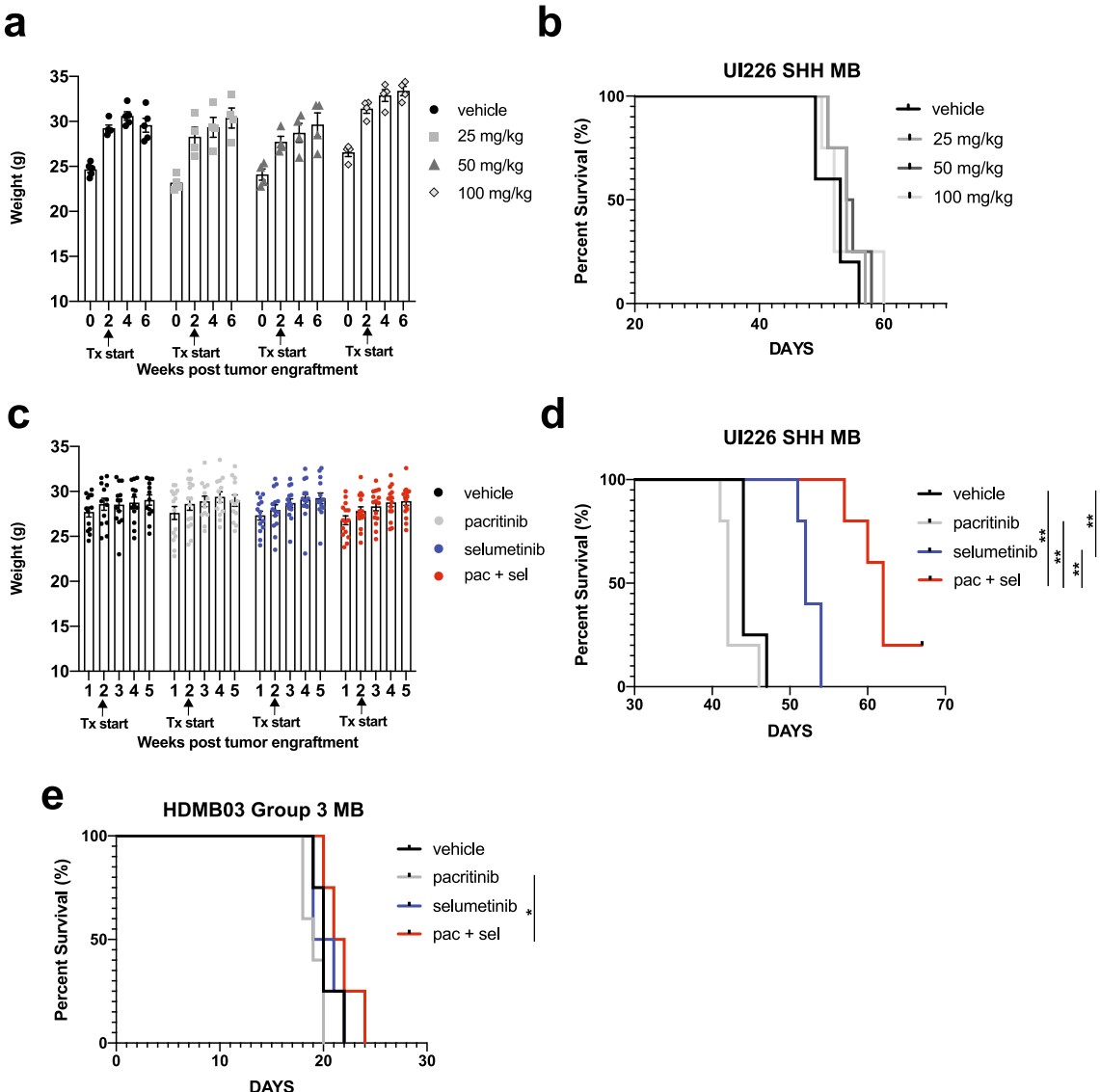

**Fig. 6 MEK + JAK/STAT3 inhibition increases survival of NOD SCID mice transplanted with UI226 SHH MB cells. a** NOD SCID mouse weights over time following vehicle control ($N = 5$), 25 mg/kg ($N = 4$), 50 mg/kg ($N = 4$) or 100 mg/kg ($N = 4$) pacritinib treatment. Arrows denote initiation of drug treatment at day 14 following injection of UI226 tumor cells. Bars: SEM. **b** Kaplan–Meier curves following transplantation of NOD SCID mice with $2.5 \times 10^5$ UI226 SHH MB cells and treated with vehicle control ($N = 5$), 25 mg/kg ($N = 4$), 50 mg/kg ($N = 4$) or 100 mg/kg ($N = 4$) pacritinib. $P$-value was determined using the log-rank method. Treatment was initiated 14 days following UI226 tumor cell injection. Animals were treated twice daily, 5 days a week by oral gavage, with a 2-day drug holiday on weekends until they reached endpoint. **c** NOD SCID mouse weights over time following vehicle ($N = 13$) (black), pacritinib only ($N = 15$) (gray), selumetinib only ($N = 14$) (blue), or pacritinib + selumetinib ($N = 15$) (red) combination drug treatment. Arrows denote initiation of drug treatment at day 14 following injection of UI226 tumor cells. Bars: SEM. **d** Kaplan–Meier curves following transplantation of NOD SCID mice with $2.5 \times 10^5$ UI226 SHH MB cells and treated with vehicle control ($N = 4$) (black), pacritinib ($N = 5$) (gray), selumetinib ($N = 5$) (blue) or pacritinib + selumetinib ($N = 5$) (red). $P$-value determined using the log-rank method. $p < 0.01$**. Vehicle vs selumetinib and vehicle vs pac/sel, $p = 0.0027$; pac vs pac/sel, $p = 0.002$; sel vs pac/sel, $p = 0.0026$. Treatment was initiated 14 days following UI226 tumor cell injection. **e** Kaplan–Meier curves following transplantation of NOD SCID mice with $1.0 \times 10^5$ HDMB03 MB cells and treated with vehicle control ($N = 4$) (black), pacritinib ($N = 5$) (gray), selumetinib ($N = 4$) (blue) or pacritinib + selumetinib ($N = 4$) (red). $P$-value determined using the log-rank method. $p < 0.05$*. pac vs pac/sel, $p = 0.0178$. Treatment was initiated 5 days following HDMB03 tumor cell injection.

We believe that this lower dose, combined with the 5-day on, 2-day off drug holiday treatment schedule, helped to mitigate long-term toxicities in our model. However, similar to the previous studies[26,42], pacritinib alone did not improve survival in our animal models. Interestingly, Jensen et al.[26] demonstrated that pacritinib was rapidly metabolized by mouse but not human liver microsomes. This rapid clearance, combined with lower overall JAK/STAT3 pathway activity in vehicle controls, may account for the lack of efficacy as a single agent in the xenografts.

Resistance to MEK inhibition due to the upregulation of STAT3 signaling has been well documented in a number of cancers including melanoma[28,29], low-grade gliomas[43], colon[44], pancreatic[45], thyroid[30], lung[46], and breast cancer[31]. Consequently, combination-based therapeutic approaches utilizing both MEK and STAT3 inhibitors have been tested and led to reduced tumor growth and increased survival in preclinical animal models[44–46]. Selumetinib has undergone extensive testing for the treatment of refractory low-grade gliomas[47,48] as well as plexiform

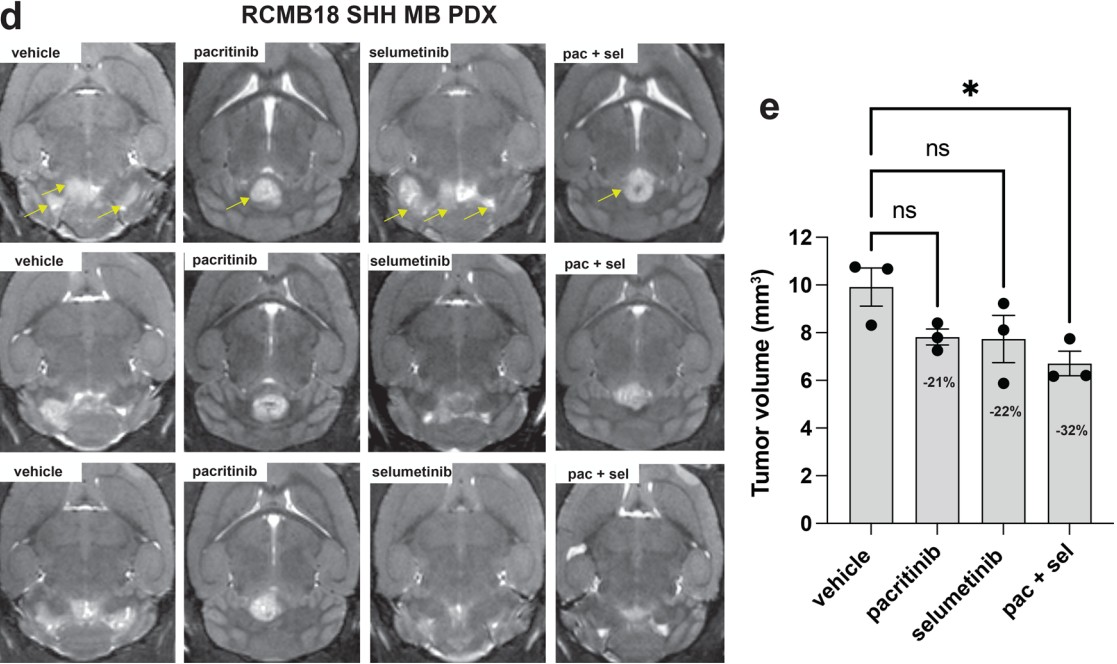

**Fig. 7 MEK + JAK/STAT3 inhibition decreases tumor growth in 2 SHH MB in vivo xenograft models.** Representative IHC images of STEM121 levels at low magnification (**a**) and higher magnification (**b**) from three independent tumors from each of the vehicle control, pacritinib, selumetinib, and combination pacritinib + selumetinib treatment groups. Scale bars: 1500 μm (**a**) and 600 μm (**b**). **c** ImageJ quantification of UI226 tumor area from STEM121 stained slides representing the vehicle control ($N = 4$), pacritinib ($N = 3$), selumetinib ($N = 4$), and combination pacritinib + selumetinib ($N = 4$) treatment groups. Results were analyzed using ANOVA and a Dunnett's test for multiple comparisons. Error bars: SEM. $p < 0.05$*. vehicle vs pac/sel, $p = 0.0105$. **d** Representative MRI images from three independent RCMB18 tumors representing the vehicle control, pacritinib, selumetinib, and combination pacritinib + selumetinib treatment groups. Arrows denote individual or multiple tumor lobes in representative images from each treatment group. **e** ImageJ quantification of RCMB18 tumor volume from compiled MRI images representing the vehicle control ($N = 3$), pacritinib ($N = 3$), selumetinib ($N = 3$), and combination pacritinib + selumetinib ($N = 3$) treatment groups. Results were analyzed using ANOVA and a Dunnett's test for multiple comparisons. Error bars: SEM. $p < 0.05$*. vehicle vs pac/sel, $p = 0.0308$.

neurofibromas[49,50], and has shown very good efficacy as well as excellent tolerability in both conditions. The MEK1/2 inhibitor trametinib is approved for adult BRAF V600E mutant cancers (ie. NCT01682083) and is also currently undergoing testing in both refractory low-grade gliomas and plexiform neurofibromas (NCT03363217, NCT02124772)[51,52]. Both MEK inhibitors have

excellent toxicity profiles in children. However, our results as well as previous studies in other cancers do raise the possibility that upregulation of other signaling pathways, particularly JAK/STAT3 activation, could potentially compensate for MEK inhibition following treatment of both diseases. Results of a recent retrospective centrally reviewed multi-center study underscore the need to

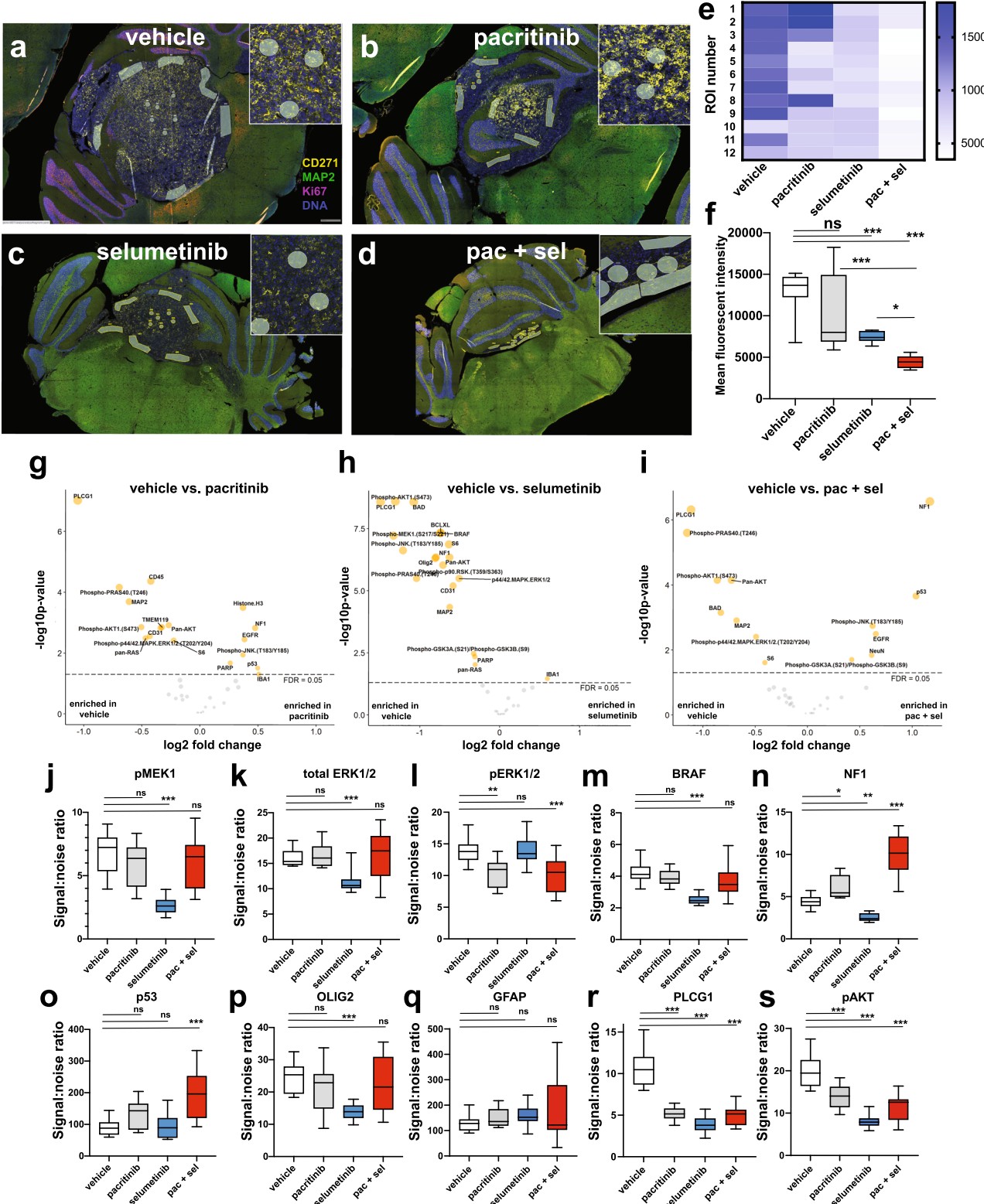

further understand the molecular mechanisms leading to rapid tumor progression in some patients following MEK inhibitor treatment[53]. Thus, more pre-clinical studies with additional brain tumor models are necessary to determine the appropriate timing for drug delivery and to more thoroughly assess sustained responses long-term. Interestingly, recent work in Group 3 MB demonstrated increased JAK/STAT3 pathway activity in chemoresistant cell lines[54]. While pacritinib treatment did not improve survival in our

Group 3 MB xenograft model, this does not exclude the possibility that treatment of chemoresistant Group 3 MB cells with JAK/STAT3 inhibitors could further improve survival in vivo.

In addition to JAK/STAT3 signaling, our RNA-seq data also suggested that the TNFα/NFκB pathway could also compensate for selumetinib treatment in SHH MB xenografts. While less widely studied than STAT3 compensation, elevated expression of TNFα has also been shown to confer resistance in response to

**Fig. 8 Multiplex digital spatial profiling of 56 proteins on MEK and/or JAK/STAT3 inhibitor-treated UI226 tumors in vivo.** Representative immunofluorescent images depicting 12 ROIs from the vehicle control (**a**), pacritinib (**b**), selumetinib (**c**) and pacritinib + selumetinib (**d**) treated samples utilized for analyses of 56 different proteins. Samples were stained for CD271 (yellow), MAP2 (green), Ki67 (pink), and Syto13 (blue) for tumor visualization. **e** Heat map depicting CD271 levels across 12 ROIs in vehicle control, pacritinib, selumetinib, and pacritinib + selumetinib treated samples. **f** Boxplots depicting quantification of CD271 levels by mean fluorescent intensity across vehicle control, pacritinib, selumetinib and pacritinib + selumetinib treated samples. Bars represent minimum and maximum counts. Significance was determined using ANOVA and a Tukey's test for multiple comparisons. $p < 0.05$*, $p < 0.001$*** For vehicle vs sel, vehicle vs pac/sel and pac vs pac/sel, $p < 0.0001$. For sel vs pac/sel, $p = 0.0393$. Volcano plots displaying significantly differentially expressed proteins (based on signal-to-noise-ratio for each target relative to negative control IgG probes comparing the vehicle control to the pacritinib (**g**), selumetinib (**h**), and pacritinib + selumetinib-treated (**i**) tumor. For each volcano plot, significance of a specific protein was determined using a two-tailed t-test. FDR: false discovery rate. Of note, 36 of the 56 proteins reached threshold levels based on signal-to-noise ratio and were used for further analyses. **j–s** Boxplots depicting select differentially expressed proteins based on signal-to-noise-ratio in vehicle control, pacritinib, selumetinib, and pacritinib + selumetinib treated samples. Bars represent minimum and maximum counts. ns: not significant. Significance was determined using ANOVA and a Dunnett's test for multiple comparisons. $p < 0.05$*, $p < 0.01$**, $p < 0.001$***. pMEK: vehicle vs sel, $p < 0.0001$; total ERK1/2: vehicle vs sel, $p < 0.0001$; pERK1/2: vehicle vs pac, $p = 0.0023$; vehicle vs pac/sel, $p = 0.001$; BRAF: vehicle vs sel, $p < 0.0001$; NF1: vehicle vs pac, $p = 0.0116$; vehicle vs sel, $p = 0.0070$; vehicle vs pac/sel, $p < 0.0001$; P53: vehicle vs pac/sel, $p < 0.0001$; OLIG2: vehicle vs sel, $p = 0.0004$; PLCG1: vehicle vs all groups, $p < 0.0001$; pAKT: vehicle vs all groups, $p < 0.0001$.

MEK inhibition. For example, MEK inhibitor resistance was observed in melanoma due to the accumulation of macrophage-derived TNFα expression and consequent downstream upregulation of the transcription factor MITF, which plays an oncogenic role in melanoma[55]. Elevation of TNFα expression in response to MEK inhibition has also been observed in lung cancer[56]. Finally, NFκB inhibition was shown to reduce MB cell growth in multiple cell lines[57]. These studies, along with our RNA-seq data, indicate that MEK inhibition in combination with TNFα /NFκB pathway inhibition should also be tested in SHH MB models.

The utilization of multiplex spatial profiling to assess the impact of treatment on xenograft models opens up new opportunities to comprehensively assess adaptive mechanisms and sustained drug responsiveness in situ. However, one limitation is the restricted number of relevant and validated antibody panels available for the spatial proteomics analyses. We were able to evaluate changes in proteins associated with neural differentiation, cell death, MAPK activity, and PI3K activity following long-term combination drug treatment in vivo. While the data enabled an initial assessment of some molecular changes, a direct comparison to the RNA-seq dataset from 5-day treated tumorspheres was difficult. Discrepancies between the short-term in vitro and long-term in vivo datasets are expected; however, future studies will expand the spatial profiling beyond proteomics to include comprehensive transcriptomic analyses to assess overlap.

We have shown that the combined inhibition of MEK and JAK/STAT3 pathway activity abrogates tumor properties in 3 different MB cell models in vitro as well as in 2 in vivo models of SHH MB. Our study reveals new insight into the altered molecular mechanisms associated with selumetinib treatment and warrants further investigation into dual MEK and JAK/STAT3 inhibition as a novel combinatory therapeutic strategy for SHH MB.

## Methods

**Cell culture and drug treatment.** UI226 (SHH MB as analyzed by NanoString[58]) cells were a kind gift from Dr. Timothy Ryken (Dartmouth-Hitchcock Medical Center, New Hampshire, USA) and have been adapted to cell culture in StemPro media as described[23–25]. Daoy cells were purchased from ATCC and cultured as described previously[23,24]. The HDMB03 Group 3 MB cell line was obtained from Dr. Till Milde[59]. All cell lines have been authenticated by STR profiling (ATCC). Tumorspheres were also propagated directly from Ptch+/−:p53+/− mouse SHH MB tumors[60]. Tumors were dissociated and Ptch+/−:p53+/− cells were grown and expanded in a NeuroCult Basal Medium (Mouse & Rat) (Stem Cell Technologies) supplemented with 7.5% BSA (1000X), 100X GlutaMAX (ThermoFisher, Scientific) in a final working solution consisting of 100X N2, 50X B27 (ThermoFisher Scientific), 10 ng/ml EGF (Sigma), 10 ng/ml bFGF (Sigma), and 2 μg/ml heparin (Sigma) as previously described[37]. Tumorsphere assays were carried out in stem cell enriched conditions as described previously[23,24]. Briefly, UI226, Daoy and Ptch+/−:p53+/− mouse SHH MB cells were plated in 24-well ultra-low attachment plates at optimal densities of 2500 cells/well (UI226), 10,000 cells/well

(Ptch+/−:p53+/− cells) and 10,000 cells/well (Daoy) and then grown for 5 days at 37 °C/ 5% CO₂. Pacritinib was added at 10, 25, 50, 100, 250 nM to cells while AZD1480 was added at 0.1, 0.25, 0.5, 1.0 and 2.0 μM. Tumorsphere numbers were counted and compared to DMSO controls. For extended selumetinib treatment and subsequent evaluation of pSTAT3 levels in vitro, Daoy cells were cultured in 1.0 μM selumetinib in stem cell enriched conditions over 3 passages.

**Cell sorting and RNA-sequencing.** UI226 tumorspheres were grown in stem cell-enriched conditions as previously described[24]. Tumorspheres were dissociated and $2.5 \times 10^5$ cells were orthotopically injected into the cerebellum of NOD SCID mice. Following 14 days of tumor growth, animals were randomly divided into two groups with one receiving selumetinib (75 mg/kg) and the other receiving the 0.5% hydroxypropyl methyl cellulose, 0.1% polysorbate 80 vehicle control via oral gavage on a 5-day on, 2-day off cycle. Upon reaching endpoint, tumors were resected and human cells were isolated by fluorescent activated cell sorting (FACS) using a human leukocyte antigen (HLA) antibody on a MoFlo XDP Cell Sorter (Beckman Coulter). RNA was extracted ($N = 3$ tumors for control and selumetinib-treated xenografts) using the Norgen RNA extraction kit (Norgen Biotek) according to manufacturer's instructions. Samples were sent for sequencing and analysis to StemCore labs and the Ottawa Bioinformatics Core Facility (OHRI) as previously described[24]. RNA-seq was performed comparing sorted control and selumetinib-treated tumor cells from primary xenografts.

RNA-seq reads were mapped independently to the human reference genome (GRCh38 assembly) guided by transcripts from GENCODE release 23, and to the mouse reference genome (GRCm38 assembly) guided by transcripts from GENCODE release M12, using hisat2 2.1.0[61]. XenofilteR[62] was run to deconvolute reads of human origin from those of mouse origin, and mapped human reads were assigned to genes from GENCODE release 23 using featureCounts[63]. Data were loaded into R and the gene/count matrix was filtered to retain only genes with 5 or more mapped reads in two or more samples. Differential expression was assessed using DESeq2 (v1.20.0)[64], comparing the two conditions (vehicle control and selumetinib-treated). Expression differences were calculated using the DESeq results function. Lists of significantly differentially expressed genes were identified using a q-value (Benjamini–Hochberg corrected p-value) cut-off of 0.05, which coincides with a p-value of 0.00168.

For analyses of drug-treated tumorspheres, transcripts from tumorsphere samples were quantified with salmon v.1.4.0[65] against an index built from the GENCODE v35 assembly with inclusion of genomic decoy sequences. Data were loaded into R using the tximport library[66] and the gene/count matrix was filtered to retain only genes with five or more mapped reads in two or more samples. Differential expression was assessed using DESeq2 v1.30.1; expression differences were calculated using the DESeq2 lfcShrink() function, applying the apeglm method (v 1.12.0)[67]. Multiple testing correction was performed using the Benjamini–Hochberg method, and lists of significantly DE genes were identified using a q-value (i.e. a corrected p value) cut-off of 0.05.

**Gene set enrichment analysis.** GSEA[68] was run using pre-ranked gene lists (the GSEA_preranked method) to explore enrichment of pathways and functional classes in gene expression changes following selumetinib or selumetinib/pacritinib treatment of xenografts and tumorspheres. The DE gene lists were filtered to retain protein-coding genes, and ranked by "-log10(p-value) * sign(fold change)". GSEA was run using the MSigDb (v6.1)[69] collections H (hallmark gene sets), C4:CP (canonical pathways), C6 (oncogenic signatures), and C8 (cell type signatures). Enrichment maps[70] were generated in Cytoscape (v3.8.2) from GSEA results, following the protocol described in Reimand et al., 2019[71]. Enriched gene sets with a q value <0.005 were retained and clustered using the AutoAnnotate tool; figures

were edited to remove unclustered gene sets, and small clusters unrelated to the cell types being studied.

**Medulloblastoma patient sample transcriptome and proteomics data**. Hierarchical clustering was performed based on the JAK/STAT signaling cascade or the entire JAK/STAT gene set from KEGG and used to determine subgroup affiliation of 763 subtyped primary MB samples previously profiled on the Affymetrix Gene 1.1ST arrays[5] and normalized using the RMA method (GSE85217). Z-scores were used to generate a signature median across subgroups. The clustering was performed in the R2: Genomics Platform (r2.amc.nl). For proteomic analyses, ProTrack[72] was used to retrieve and visualize protein data from 218 tumors across several histological classes of childhood brain cancers[35]. JAK/STAT3 signaling signature genes were chosen from our Hallmark IL6 JAK/STAT3 signaling signature gene set in Supplementary Data 5.

**High-Content Tumorsphere Assay**. Approximately 4000 UI226 cells were seeded into each well of a 96-well ultra-low attachment plate. All plates contained 3 technical replicates per condition and final results are based on averages from 3 biological replicates. Cells were seeded in various drug concentrations or in the presence of vehicle (DMSO at 0.0025%). For AZD1480 or pacritinib, 15 serially diluted concentrations ranging from 10.0 μM to 0.60 nM were dispensed in triplicate across the plate. Fifteen serially diluted concentrations ranging from 0.002–40 nM were dispensed in triplicate for the chemotherapeutic agent vincristine. Cells were incubated at 37 °C for 5 days. Following incubation, tumorspheres were fixed and stained with a 4% formaldehyde + 300 ng/mL Hoechst 33342 (Thermo Scientific) solution at 4 °C overnight (O/N).

The following day, high-content imaging microscopy was performed using a Cytation 3 imager (BioTek, Winooski, Vermont) equipped with a 16-bit gray scale 1.25 megapixel Sony charge-coupled device camera and a motorized stage. Using a 4× objective, 30 partially overlapping images were acquired per well. Images were digitally aligned to produce a single composite image for analysis.

Automated software (Gen5) was employed to identify and score tumorsphere sizes. Sizes were determined using the ellipsoid equation and measured in μm. Perimeter wells were excluded from analyses to mitigate edge-associated artifacts. To identify and digitally mask tumorspheres, Hoechst detection threshold sensitivity was set to 10,000, and only spheres containing diameters between 45 and 360 μm with circularities greater than 0.3 were scored. Tumorsphere size data were imported into Prism and optimal concentration ranges were identified for further functional studies. To validate selected concentration ranges, Fiji/ImageJ[73] was used to evaluate individual tumorsphere size in multiple wells per treatment group. Results were displayed as cumulative frequency distribution of tumorsphere area. Only tumorspheres greater than 25 μm were analyzed.

**Immunohistochemistry**. Xenograft tumors were sectioned following formalin-fixation and paraffin-embedding. Slides were de-parafinized and re-hydrated in xylene and ethanol gradients followed by antigen retrieval in sodium citrate at 95–100 °C. Slides were washed and then treated for endogenous peroxidases for 10 min. Slides were blocked in 3% lamb serum for 30 min at room temperature (RT) followed by O/N incubation at 4 °C with primary antibody (Supplementary Table 1) prepared in 1% lamb serum/PBS. The following day, slides were first incubated in biotinylated secondary antibodies for 2 h, followed by streptavidin/HRP for 30 min at RT. Slides were then developed with DAB substrate (Sigma), counterstained with hematoxylin, and mounted in Permount (ThermoFisher Scientific).

**Immunoblotting**. UI226, Daoy, and Ptch+/−:p53+/− mouse SHH MB tumorspheres were grown in culture and treated with pacritinib or AZD1480 as described above. Proteins were loaded onto 10–12% Tris-glycine gels and resolved by SDS-PAGE. Protein was transferred on to nitrocellulose membranes, blocked with 5% milk, and incubated O/N in primary antibodies described in Supplementary Table 1. Secondary antibodies conjugated to HRP were added the following day for 1 h at RT. Signal detection was performed using SuperSignal West Pico. Images were captured with a Fusion FX Vilber Lourmat chemiluminescent imaging system (Marne-la-Vallee cedex 3, France).

**High-content tumorsphere drug synergy assay**. To evaluate in vitro drug synergy, 4000 UI226 cells/well were seeded into 25 wells (a five-by-five matrix) of a 96-well ultra-low attachment plate. Serially diluted concentrations of selumetinib were dispensed in the y-direction while serially diluted concentrations of AZD1480 or pacritinib were dispensed in the X-direction such that the uppermost left well and bottom most right well contained the most dilute and concentrated drug combinations, respectively. Cells were incubated for 5 days in the presence of drug combinations, fixed, counterstained, imaged, and analyzed in Prism. Each biological replicate employed 3 technical replicates and results were based on a total of three biological replicates. Data were averaged, normalized, and imported into Combenefit[74] software for synergy calculations. Data were analyzed using the classical Loewe synergy model and values presented in matrices represent the percentage of tumorsphere size reduction that is greater than predicted by an additive model.

**Migration assays**. Migration assays were performed as previously described[24]. Briefly, Daoy, UI226, and Ptch+/−:p53+/− MB cells were seeded in round bottom 96-well ultra-low attachment plates at $1 \times 10^4$ cells/well, $4 \times 10^3$ cells, and $5 \times 10^3$ cells, respectively. Individual aggregates or spheroids were formed after 4 days, and overlain with a Type I collagen mixture (Collagen type I, DMEM, and NaOH). Once the collagen solidified, DMSO, single agent or combination drug therapies were added. Migration was assessed after 3 days in culture by subtracting the diameter measured at day 0 from the diameter measured at day 3.

**Annexin V staining**. The Annexin V Apoptosis Detection Kit (BD Biosciences) was used to evaluate cell death as previously described[24]. Dissociated UI226 SHH MB tumorspheres grown in stem cell conditions and treated with pacritinib + selumetinib or AZD1480+ selumetinib were stained with Annexin V, followed by 7AAD. Samples were run on a MoFlo XDP cell sorter (Beckman Coulter, Mississauga, ON, Canada) and analysis was performed using Kaluza software (Beckman Coulter).

**Intracerebellar transplantation and drug treatment**. The University of Manitoba Animal Care Committee approved all procedures and protocols. UI226 SHH or HDMB03 Group 3 MB cells were injected into NOD SCID mice as previously described[20,23,24,40,75,76]. Briefly, UI226 or HDMB03 tumorspheres were dissociated and either $2.5 \times 10^5$ cells (UI226) or $1.0 \times 10^5$ cells (HDMB03) were injected into the cerebellum of 7–9 week old NOD SCID mice. RCMB18 is a MYCN amplified, TP53 mutated SHH MB PDX model[15]. RCMB18 cells were dissociated and $2.5 \times 10^5$ cells were injected into the cerebellum of NOD SCID IL2Rg null (NSG) mice. Tumors were propagated and then transplanted into secondary recipients for drug treatment. Following tumor engraftment, animals were randomly separated to either receive vehicle control, selumetinib (37.5 mg/kg), pacritinib (50 mg/kg), or pacritinib + selumetinib (50 mg/kg pacritinib and 37.5 mg/kg selumetinib) twice daily 5 days a week by oral gavage, with a 2 day drug holiday on weekends. For UI226 and RCMB18, treatment was initiated 14 days following tumor cell injection. For HDMB03, treatment was initiated 5 days following tumor cell injection. Animals were treated until study endpoint. All drugs were resuspended in sterile 0.5% hydroxypropyl methyl cellulose + 0.1% polysorbate 80. RCMB18 tumor MRI was performed on a MR Solutions cryogen free FlexiScan 7T system (MR Solutions, Guildford, Surrey, UK). Endpoint was determined as a 20% weight loss from peak, lethargy, and ruffled fur. Following endpoint, animals were perfused and brains were extracted for histological analysis as previously described[23,40].

For tumor size calculations in both IHC ($N = 4$ samples per treatment group) and MRI images ($N = 3$ mice per treatment group), the ImageJ freehand tool was used to determine area and volume respectively. STEM121 staining was used to demarcate the human tumor cell margins in IHC images. Tumor tissue, consisting of 1 or multiple lobes, in the cerebellum was visually identified on MRI images and used to delineate tumor size using the freehand tool. For each sample, 18 serial sections (300 μm thickness) were obtained by MRI and volume calculated for each slice containing visible tumor tissue. The sum from all the slices containing tumor tissue was used to calculate an overall tumor volume.

**Digital spatial profiling on drug-treated xenografted medulloblastoma samples**. Four representative slides from FFPE vehicle, pacritinib, selumetinib and pacritinib + selumetinib drug-treated xenografted tumor samples were processed following the GeoMx™ DSP slide preparation user manual (MAN-10087-04). Slides were baked at 60 °C for at least 1 h, and then rehydrated and blocked using NanoString blocking buffer for 1 h. CD271/NGFR-594 (Abcam, ab221212), Ki67-647 (Cell Signaling Technology, 9027S), and MAP2-532 (Novus Bio, NBP1-92711AF532) were added to the sections along with the NanoString protein antibody cocktail that included the available Neuro core panel, a MAPK panel, a PI3K panel and a cell death panel (NanoString Technologies). For O/N incubation at 4 °C. The slides were washed and stained with Syto13 (ThermoFisher, S7575) for 5 min before loading onto the GeoMx™ DSP microscope (NanoString Technologies). Fluorescent images were scanned at 20X and ROIs in the tumor core and along the tumor boarder were selected. Oligos from antibodies were cleaved and collected into 96-well plates, hybridized with NanoString barcodes O/N and then read with an nCounter machine the following day. The digital counts of each antibody for each ROI were generated for data analyses. Per manufacturer's recommendations, the data were normalized by scaling to the negative control IgG probes, which reflect the rate at which antibodies bind to each region, called signal to noise ratio (SNR). The SNR of each protein target from vehicle, pacritinib, selumetinib, and pacritinib + selumetinib samples was calculated. Significance was determined using paired t-tests (volcano plots) or a one-way ANOVA and a Dunnett's test for multiple comparisons (box plots). The HDMB03 PQR620-treated samples[40] underwent the same procedures and workflow; however, CD271 was not included as a morphology marker.

**Fluorescence intensity measurements of CD271**. CD271/NGFR-594 (Abcam, ab221212) was used as a morphology marker in guiding ROI selections as stated above. TIFF images from all ROIs were downloaded from the GeoMx™ DSP instrument and used to measure the mean fluorescence intensity of CD271 using

Image J. CD271 levels in all samples were analyzed by one-way ANOVA and Tukey's test for multiple comparisons.

**Statistics and reproducibility**. Prism 8.0 software (GraphPad) was used for all statistical analyses. The Log-rank (Mantel–Cox) test was utilized to evaluate xenograft survival data. ANOVA followed by a Tukey's test for multiple comparisons were used to assess in vitro assays involving combination drug treatment. Single drug experiments were examined using one-way ANOVA and a Dunnett's test for multiple comparisons. Brown–Forsythe tests were performed to assess homogeneity of variances. Additional tumorsphere size analysis was performed using two-sample Kolmogorov–Smirnov tests. Significance was considered at $P$ values ≤0.05. All data reported as ± SEM.

**Reporting summary**. Further information on research design is available in the Nature Research Reporting Summary linked to this article.

## Data availability

RNA-seq data have all been deposited into GEO under the accession code GSE165960. All supportive data corresponding to the GEO files are displayed in Figs. 1 and 5, as well as Supplementary Fig. 1 and Supplementary Data 1–7. Additional publicly available datasets utilized include GSE85217 and the Petralia et al.[35] proteomics data on ProTrack data portal (http://pbt.cptac-data-view.org/). Original, unprocessed immunoblots associated with Fig. 3c–e and Supplementary Figs. 1b, d and 3a–c can be found in Supplementary Fig. 7a, b online. All source data underlying the graphs and charts presented in the main figures can be found in Supplementary Data 11.

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

## Acknowledgements

We thank Agnes Fresnoza and Shawn Blum from Vet Services at the University of Manitoba for outstanding technical support during the course of our animal studies. We thank Dr. Mike Jackson from the Small Animal and Materials Imaging Core Facility, University of Manitoba, for performing MRI. This work was funded by the Canada Research Chairs Tier II Program, and operating grants from the Canadian Institutes of Health Research, The Rally Foundation for Childhood Cancer Research, and the CancerCare Manitoba Foundation (all to TEWO). J.Z. was supported by a joint Postdoctoral Fellowship from the CancerCare Manitoba Foundation, the Children's Hospital Research Institute of Manitoba and Research Manitoba. S.B. was supported by the Research Manitoba-University of Manitoba WM Ross Scholarship-CancerCare Manitoba Foundation Master's Studentship Award. B.J.G. was supported by a William Donald Nash Brain Tumor Foundation of Canada Fellowship.

## Author contributions

J.Z., S.B., and T.E.W.O. were primarily responsible for conception and design of experiments, data collection, assembly and interpretation of data, manuscript writing, and final approval. Acquisition of data, data assembly, and interpretation: B.J.G., G.M.S., L.C.M., V.G., L.L., S.C., R.K., C.H., J.A.C., C.J.P., Y.L., J.G., C.N., O.S., R.W.R., and V.R. Additional manuscript writing and review: J.Z., S.B., L.C.M., and T.E.W.O.

## Competing interests

The authors declare no competing interests.
