## [Peer Review File · Communications Biology]

Reviewers' comments:

Reviewer #1 (Remarks to the Author):

Combine MEK and JAK/STAT3 pathway inhibition effectively decreases medulloblastoma tumor progression

In this manuscript, which is a follow-up of their previous publication on the efficacy of the MEK inhibitor selumetinib in SHH medulloblastomas, Zagozewski et al., use RNA sequencing on selumetinib-treated tumors to investigate compensatory mechanisms that compensate for MEK inhibition leading to therapeutic resistance and tumor recurrence. They identify the JAK/STAT3 pathway as being upregulated in selumetinib-treated tumors and investigate combinatorial therapeutic strategies with JAK/STAT3 inhibitors and selumetinib for the SHH subclass of MBs.

The study is interesting and follows the current thought that there are compensatory mechanisms triggered upon single agent therapies and highlight the need for combinatorial treatment. The study presented here uses a variety of technologies and new platforms to study the effect of combinatorial treatments with selumetinib with JAK/STAT3 inhibitors. Some of the studies ask the right questions, however the experimental designs have some flaws, which if addressed would make the manuscript much more scientifically rigorous.

1. Representative images of histology from the brains of treated and untreated animals are needed for assessment of tumors (e.g burden, invasion etc..) at a similar time point as the tumors being sorted. What is the time point following xenografts that these tumors were collected? Where is this time point relative to the Kaplan Meier survival of these mice (i.e what time point was used for end point in days and was there any variability for the 3 mice per group?).
2. Figure 1F, scale bars are missing for the IHC sections
3. The GSEA and gene sets are difficult to navigate as currently represented. Figure panels showing pathway analyses (IPA, GO terms etc..) would make the data more visually represented. As well there should be some stats for the top up/down regulated pathways to further support the rationale for focusing on the JAK/STAT pathway. To further support focusing on the JAK/STAT pathway and in particular STAT3 activation, can the authors show some in vitro data such as western blots following selumetinib treatment as well as down regulation of the target pathways/genes in figure 1e?
4. How was the scoring in Figure 1g performed? The graph as currently represented is just a percentage and could be a somewhat biased analysis based on the section being scored etc. The number of positive v/s negative cells should be counted in at least 3-5 sections from each brain. Thresholds can be set for level of staining.
5. Figure 2 is interesting but is not really relevant as a main figure, could be supplementary material to Figure 1 following the identification of the JAK/STAT3 pathway as one of the main pathways upregulated following selumetinib treatment. The author's conclusion that the lower expression of JAK/STAT3 regulated gene signatures being lowest in the SHH group hence supporting the finding that it goes up following treatment is purely conjecture since they did not test all of the sub groups. This statement is best removed and the figure shifted to supporting information.
6. The authors use tumor size to screen for pacritinib and AZD 1480. While this is an acceptable method for determining drug effect, the scoring methodology and graphical representation and determination of EC50 are not statistically accurate. They should either count the total number of spheres remaining after treatment or do a proper EC50 estimation by using higher dose response concentration values to reach close to 100% effect. The graphs shown in Figures 2C and 2D never reach full response with the pacritinib effect levelling off at 70% and the AZD1480 even higher at greater than 80%. Not sure how these EC50 values were therefore determined in a pharmacologically and statistically relevant manner. Were the remaining tumorspheres assessed for live/dead cells? Adding an agent such as alamarBlue at the end point of this experiment would also help supplement this data as well as enable more accurate assessment of EC50. Otherwise, please replot as bar graphs

to show reduction in tumorsphere size at relevant doses vs EC50 curves.

7. Please include a western blot panel showing upregulation of pSTAT3 in vitro following selumetinib treatment in vitro

8. For figure 4, it is also unclear why tumor size is the scoring parameter vs viability for the synergy studies. It is not convincing whether the reduction in tumour size is due to cell death or cell cycle arrest. The authors do go into a bit more granular assessment of viability and cell death in the supplementary data, however this really detracts from the main figure and is quite difficult to follow with multiple panels etc. It would be much easier to interpret the data if viability was the primary readout to begin with in the synergy studies

9. All of the data in Figures 3 and 4 should be represented as normalized values to DMSO controls

10. The data in Figure 5 is misplaced as a main figure. The reviewer does not see any direct relevance of this analysis. It can be shifted as supplemental information.

11. Total Duration of treatment is not clear for Figure 6a in vivo studies, please include in legend or directly on figure panel.

12. Please include scale bars in Figure 7

13. Number of animals used for the KM studies in figure 7 are very few for real KM studies

14. How is tumor size/burden determined? Authors need to provide details for methodology used for graphs in 7c and 7e

15. Significant toxicity has been previously reported with long term pacritinib treatment in mice vs humans, it is intriguing that the authors do not observe this in their model. They should elaborate on this a bit more in the discussion and also as per point 11, specify the total treatment duration.

16. Figure 8E, what is the significance of ROIs between selumetinib or pacritinib treatment and the combinatorial treatment?

17. Why was pSTAT3 assessment not performed in the ROI proteomic studies, would be more relevant to the premise of this study.

Reviewer #2 (Remarks to the Author):

In this paper, the authors identified the treatment by MEK inhibitor (selumetinib) decreased SHH MB growth while extending survival in mouse models. But the mice ultimately died from disease progression. Through RNA sequencing on selumetinib-treated orthotopic xenografts, the authors identified molecular pathways that compensate for MEK inhibition. They focused on JAK/STAT3 pathway due to its increased activation in selumetinib-treated tumors. The novel combination of the MEK and the JAK/STAT3 pathway inhibitors was reported to reduce tumor growth in two xenograft models and enhances survival. Multiplex spatial profiling of proteins in drug-treated xenografts reveals shifted molecular dependencies and compensatory changes following combination drug treatment. Taken together, this paper provided evidence showing MEK and JAK/STAT3 inhibition is a novel combinatory therapeutic strategy for SHH MB.

This work is rigor and nicely illustrated. The idea of this work is initiated by the authors' own original findings. It is of interest to me and may be interesting to the wider field since the authors showed the potential of MEK and JAK/STAT3 combinatory therapy in the future, which may benefit the MB patients. Meanwhile, the digital spatial profiling reveals unique changes in proteins associated with MAPK signaling and cell death following combination therapy treatment, with which may inspire further studies and identify novel therapy target for all the subtypes of MB. Although this paper is of high quality, I still have concerns.

Major concerns:

1. Some occasional grammar mistakes could be seen in the text. Careful language editing by native speakers is recommended.

2. The authors established in vitro model of tumorspheres and PDX model for in vivo study. However, all the study models are SHH MB, the findings of this paper are not verified in the other

subtypes of MB such as WNT, Group 3 and Group 4. While Group 3 and Group 4 MBs are well known to have much poorer prognostic and high recurrence rate. Thus, the title is recommended to be modified as "Combined MEK and JAK/STAT3 pathway inhibition effectively decreases SHH medulloblastoma tumor progression"

3. The toxicity and side effect of the dual MEK and JAK/STAT3 inhibition is not tested in this study. Discussion about this issue is necessary.

4. RNA sequencing for SHH MB in tumorspheres and digital spatial profiling for PDX models reveal significant molecular changes in SHH MB tumorspheres following dual MEK and JAK/STAT3 inhibition therapy. However, overlap of molecular changes between the in vivo and in vitro study is barely seen. Could you please explain this question in the discussion?

5. Regarding the MRI image in figure 7d. The authors declare that "combination drug treatment of RCMB18 SHH MB patient-derived xenografts (PDX) in a small cohort of NOD-SCID IL2Rg null (NSG) mice significantly reduced tumor growth, as only single well-encapsulated nodules were observed relative to vehicle controls." But it seems that in the groups of pacritinib and pac+sel group, the sections are similar, but in the vehicle and sel group the section is higher, the same section of MRI is necessary. Additionally, in the pac group, the tumor is inerratic and encapsulated as well. Discussion is in need.

Decision: Major revision

Reviewer #3 (Remarks to the Author):

Zagozewski et al examine the effects of MEK and JAK/STAT inhibition on SHH-medulloblastoma tumor growth, both in vitro and in vivo. By examining RNA-seq data of MEKi-treated mice, the authors identify the JAK/STAT pathway as upregulated in treated animals compared with control animals. To further examine the role of JAK/STAT in MB growth, the authors first examine the effect of two JAK/STAT inhibitors on 2 human MB cell lines and 1 mouse MB cell line. After identifying pacritinib as effective in vitro, the authors perform combination treatment studies in vivo and show that lifespan of the animals is further extended, although the majority of animals eventually succumb to disease.

The manuscript flows nicely and in a logical manner, making the course of experiments easy to follow. It is certainly interesting how the JAK/STAT pathway is used by the tumor cells to compensate for MEK pathway inhibition, especially since this pathway isn't usually upregulated in SHH-MB patients. The low toxicity of the combination treatment also makes it a relevant therapeutic strategy and warrants future consideration.

A few specific comments/concerns:

1. It is difficult to read/understand Fig 3A. Could you please make this figure larger and explain a bit more in the legends what the point of the figure is. Also for some of the legends, you do not state the statistical test used (For example Fig3G-L, but also others).

2. I find it puzzling that AZD1480 seems to obliterate pSTAT3 at all concentrations tested (from the WB), also when compared with the pacritinib treated WB, where this effect is concentration dependent. However, the effect of AZD1480 treatment on tumorsphere size does not correspond to the levels of pSTAT3. Could it be that tumorsphere size is affected by something other than pSTAT inhibition (some off-target effect of pacritinib?) How can you reconcile these differences?

3. Along those same lines – are you sure that the ptch;p53 mouse cells have pSTAT at basal levels? Perhaps it would be useful to show a similar WB for those cells, assuming you have material/antibodies.

4. In figure 7, I'm surprised you still have neurological signs of tumor formation with such small tumors (pac+sel treated). Are these end point tumors? If not, when were these mice sacrificed in relation to treatment/tumor growth? Does this mean the tumor grows back after initially shrinking? For Figure 7, I also think estimating tumor volume (not area) would be more helpful, since area relies on where you cut in FFPE sections. Can you use the FFPE blocks to estimate tumor volume? Same for the RCMB18 – can you estimate tumor volume on the MRI instead of area? For these images, it would also be helpful for those not used to looking at MRIs to mark on the figures where the tumor is located.

Reviewer #1:

In this manuscript, which is a follow-up of their previous publication on the efficacy of the MEK inhibitor selumetinib in SHH medulloblastomas, Zagozewski et al., use RNA sequencing on selumetinib-treated tumors to investigate compensatory mechanisms that compensate for MEK inhibition leading to therapeutic resistance and tumor recurrence. They identify the JAK/STAT3 pathway as being upregulated in selumetinib-treated tumors and investigate combinatorial therapeutic strategies with JAK/STAT3 inhibitors and selumetinib for the SHH subclass of MBs.

The study is interested and follows the current thought that there are compensatory mechanisms triggered upon single agent therapies and highlight the need for combinatorial treatment. The study presented here uses a variety of technologies and new platforms to study the effect of combinatorial treatments with selumetinib with JAK/STAT3 inhibitors. Some of the studies ask the right questions, however the experimental designs have some flaws, which if addressed would make the manuscript much more scientifically rigorous.

Response: We thank the reviewer for the insightful comments and appreciate that the reviewer found our work to be of interest.

1: Representative images of histology from the brains of treated and untreated animals are needed for assessment of tumors (e.g burden, invasion etc..) at a similar time point as the tumors being sorted. What is the time point following xenografts that these tumors were collected? Where is this time point relative to the Kaplan Meier survival of these mice (i.e what time point was used for end point in days and was there any variability for the 3 mice per group?).

Response #1: We apologize for the lack of detail regarding the samples taken from our previously published paper (Liang et al., Cancer Research, 2018). The published Kaplan Meier curves from this study are shown for reference below in Rebuttal Figure 1. We have now added representative H&E images of the control and selumetinib-treated tumors utilized for the RNA sequencing studies presented in updated Figure 1b of the revised manuscript.

Rebuttal Figure 1: MRI images (A) and Kaplan Meier curves (B) comparing control vehicle and selumetinib treatment in UI226 SHH MB tumor-bearing NOD SCID mice. Taken from *Cancer Res.* 2018;78(16):4745-4759. doi:10.1158/0008-5472.CAN-18-0027

Additional details requested about the timepoints are now included on page 5 of the revised manuscript as follows:

“To identify aberrant molecular changes following selumetinib treatment, we performed RNA sequencing (RNA-seq) on sorted control (N=3 independent endpoint tumors at days 43 and 46 (2 mice) post-tumor cell injection) and selumetinib-treated (N=3 independent endpoint tumors at days 56, 58 and 59 post-tumor cell injection) UI226 SHH MB cells (recently derived from a primary

SHH MB tumor^{23,25}) from tumor xenografts previously characterized²⁴ (Fig. 1a-b). Endpoint tumors from both the control and selumetinib groups were histologically similar displaying high nuclear-to-cytoplasmic ratios and well-encapsulated margins (Fig. 1b).” H&E images displayed in Figure 1b are also shown below in Rebuttal Figure 2 for reference.

Rebuttal Figure 2: Representative H&E images of FFPE sections from control (left) and selumetinib-treated (right) UI226 MB xenografts at endpoint. Scale bars: 1500 μ m (upper) and 600 μ m (lower).

2. Figure 1F, scale bars are missing for the IHC sections

Response #2: Thank you, scale bars have now been added to each image in the figure panel.

3. The GSEA and gene sets are difficult to navigate as currently represented. Figure panels showing pathway analyses (IPA, GO terms etc..) would make the data more visually represented. As well there should be some stats for the top up/down regulated pathways to further support the rationale for focusing on the JAK/STAT pathway.

Response #3A: We thank the reviewer for the comment. As GSEA is commonly used as a display tool in manuscripts, we prefer to utilize this format. That being said, we completely agree that statistics for the top/downregulated pathways should be incorporated. We have now added tables (Supplementary Tables 2 and 3 which are also included below as Rebuttal Tables 1-2 for reference) for the top up/downregulated hallmark gene sets in selumetinib-treated xenografted tumor cells vs. controls. Of note, Supplementary Table 2 clearly demonstrates the statistical relevance of the JAK/STAT3 pathway. Moreover, in our manuscript, to further strengthen novelty and our rationale, we stated that our choice to pursue the JAK/STAT3 pathway was based on the following:

1. Very little is known about the role of the JAK/STAT3 pathway in human SHH MB tumor progression and drug resistance.
2. Several drugs targeting JAK/STAT3 signaling have been identified and are known to be brain penetrant in xenograft models including pacritinib and AZD1480.
3. Recent studies have also shown that STAT signaling activation accompanies MAPK pathway inhibition in multiple cancers thus implicating upregulated JAK/STAT3 activity as a general mechanism associated with drug resistance or long-term adaptive responses in multiple cancers.

Furthermore, we are preparing to submit a separate manuscript demonstrating the benefit of another, more potent MEK inhibitor, trametinib, in our *in vivo* models, and the RNA-sequencing results are very similar to those observed with selumetinib in Figure 1 of our current manuscript (see Rebuttal Figure 3 below). Again, top hallmark gene sets enriched in drug treated xenografts are IL6/JAK/STAT3 signaling and TNF α signaling via NF κ B (Rebuttal Figure 3), providing further support for our original *in vivo* data years later but with a different MEK inhibitor. Our goal is to publish this new study after the current one back-to-back.

Rebuttal Figure 3: GSEA demonstrating hallmark gene sets significantly ($p < 0.05$ *) enriched in xenografts treated with the MEK inhibitor trametinib.

Rebuttal Table 1: Hallmark gene sets that were significantly ($p < 0.01$) enriched in genes that are upregulated in selumetinib-treated tumors

Gene set	SIZE	ES	NES	NOM p-val	FDR q-val
HALLMARK_INTERFERON_GAMMA_RESPONSE	179	0.865013	2.762093	0	0
HALLMARK_INTERFERON_ALPHA_RESPONSE	90	0.885502	2.652913	0	0
HALLMARK_TNFA_SIGNALING_VIA_NFKB	176	0.758741	2.417747	0	0
HALLMARK_ALLOGRAFT_REJECTION	140	0.707954	2.224228	0	0
HALLMARK_INFLAMMATORY_RESPONSE	142	0.709648	2.222453	0	0
HALLMARK_IL6_JAK_STAT3_SIGNALING	69	0.676055	1.970014	0	0
HALLMARK_APOPTOSIS	145	0.586575	1.84928	0	7.92E-04
HALLMARK_IL2_STAT5_SIGNALING	168	0.578675	1.833926	0	6.93E-04
HALLMARK_COMPLEMENT	152	0.570987	1.829139	0	6.16E-04
HALLMARK_ESTROGEN_RESPONSE_EARLY	169	0.526037	1.660343	0	0.004835
HALLMARK_TGF_BETA_SIGNALING	54	0.58849	1.626272	0.005208	0.00741
HALLMARK_UV_RESPONSE_DN	135	0.512566	1.600614	0.003356	0.009477
HALLMARK_HYPOXIA	178	0.496231	1.594177	0.001095	0.009873

Rebuttal Table 2: Hallmark gene sets that were significantly ($p < 0.05$) enriched in genes that are downregulated in selumetinib-treated tumors

Gene set	SIZE	ES	NES	NOM p-val	FDR q-val
HALLMARK_OXIDATIVE_PHOSPHORYLATION	180	-0.4412	-1.8339	0	0.0094
HALLMARK_ANGIOGENESIS	29	-0.5393	-1.6222	0.0035	0.0217

To further support focusing on the JAK/STAT pathway and in particular STAT3 activation, can the authors show some *in vitro* data such as western blots following selumetinib treatment as well as down regulation of the target pathways/genes in figure 1e?

Response #3B: As stated in the manuscript, we chose *in vivo* vs. *in vitro* screen-based analyses, as we believe this will reveal pathways that may compensate for MEK inhibition in the more complex milieu of the cerebellar tumor microenvironment and reflect long-term adaptive responses. It is well known that results from *in vitro*-based studies are not always replicated *in vivo*, especially when applied to the brain tumor field.

Importantly, our Figure 1 xenograft data are based on long-term (30-40+ days) treatment as a model of compensation and potential drug resistance. Thus, it would not be appropriate to compare these results to tumorspheres in stem cell-enriched conditions over a very short 1-5 day period and expect all the same outcomes at a molecular level. As proof of principle, we have performed new experiments to show that tumorspheres treated with selumetinib start to undergo apoptotic cell death very early (Rebuttal Figure 4 below). After 1-3 days treatment, pSTAT3 levels remain constant at the lowest concentration (1 μ M); however, not surprisingly, in the dying cells exposed to high concentrations, pSTAT3 is decreasing. As such, these *in vitro* studies do not provide an adequate comparison to the *in vivo* data.

In fact, the only way to better compare the data from the long-term selumetinib-treated xenografts would be to generate selumetinib-resistant cell lines *in vitro* over several weeks/months. Even this selection process could result in entirely different molecular dependencies given the fact that the tumorspheres are in well-defined stem cell-enriched conditions. That being said, we have performed these new experiments in UI226 and Daoy cells at the lowest 1 μ M selumetinib dose. Unfortunately, the UI226 tumorspheres were very sensitive to treatment, and we could not obtain sufficient material for western blots after the first passage (Rebuttal Figure 5A below). However, we were able to pass the Daoy tumorspheres 3x and an increase in pSTAT3 was observed at all time points (Rebuttal Figure 5B below). Given the emphasis on preclinical xenograft data throughout and the fact that the combination therapy was shown to be effective in our animal model, we have not included these new results in the revised manuscript; however, if the reviewers feel as though the data should be added, we can incorporate.

Rebuttal Figure 4. Treatment with the MEK1/2 inhibitor selumetinib and the effect on pERK and pSTAT3 levels. Western blot of pERK1/2, pSTAT3 and cleaved caspase 3 levels in selumetinib-treated primary Daoy tumorspheres at day 3. Total ERK, Total STAT3 and GAPDH serve as loading controls.

A**B**
Rebuttal Figure 5. Treatment with the MEK1/2 inhibitor selumetinib results in higher levels of pSTAT3 over subsequent passage. A) Representative image of Daoy (upper) and UI226 (lower) tumorspheres following 1 passage (6 days) in either DMSO or 1 μM selumetinib. Scale bar: 300 μm. B) Western blot depicting pSTAT3 levels following selumetinib treatment of Daoy tumorspheres over 3 passages (P1-P3). Total STAT3 and GAPDH serve as loading controls.

4. How was the scoring in Figure 1g performed? The graph as currently represented is just a percentage and could be a somewhat biased analysis based on the section being scored etc. The number of positive v/s negative cells should be counted in at least 3-5 sections from each brain. Thresholds can be set for level of staining.

Response #4: We thank the reviewer for the very valuable suggestion. We have now performed quantification using QuPath in the samples that exhibited detectable pSTAT3 staining by IHC. These data are indicated below in Rebuttal Figure 6 and are now in Figure 1h of the revised manuscript. Data are presented as the number of pSTAT3+ cells relative to the total number of cells (pSTAT3+ cells and number of hematoxylin+ nuclei) in each 10x image.

Rebuttal Figure 6. Quantitative analyses of IHC pSTAT3 (Tyr705) staining in vehicle control (white) and selumetinib-treated (blue) xenografts. The proportion of pSTAT3+ cells for each sample was calculated using QuPath and expressed as the total number of pSTAT3+ cells relative to the total number of cells in each image (pSTAT3- nuclei (hematoxylin-stained) and pSTAT3+ cells). Significance was calculated using a two-tailed t-test. Error bars: SEM. $p < 0.05$ *.

5. Figure 2 is interesting but is not really relevant as a main figure, could be supplementary material to Figure 1 following the identification of the JAK/STAT3 pathway as one of the main pathways upregulated following selumetinib treatment. The author's conclusion that the lower expression of JAK/STAT3 regulated gene signatures being lowest in the SHH group hence supporting the finding that it goes up following treatment is purely conjecture since they did not

test all of the sub groups. This statement is best removed and the figure shifted to supporting information.

Response #5: We thank the reviewer for the suggestion; however, given the overall emphasis on *in vivo*/preclinical data in the manuscript, we feel that inclusion of JAK/STAT signatures at the transcript and protein levels across a large number of pediatric brain tumor samples is informative and relevant to the main figures. We believe these data should remain included; however, we have refined the concluding statement on page 6 of the revised manuscript to specifically focus on our model.

6. *They authors use tumor size to screen for pacritinib and AZD 1480. While this is an acceptable method for determining drug effect, the scoring methodology and graphical representation and determination of EC50 are not statistically accurate. They should either count the total number of spheres remaining after treatment or do a proper EC50 estimation by using higher dose response concentration values to reach close to 100% effect. The graphs shown in Figures 2C and 2D never reach full response with the pacritinib effect levelling off at 70% and the AZD1480 even higher at greater than 80%. Not sure how these EC50 values were therefore determined in a pharmacologically and statistically relevant manner. Were the remaining tumorspheres assessed for live/dead cells? Adding an agent such as alamarBlue at the end point of this experiment would also help supplement this data as well as enable more accurate assessment of EC50. Otherwise, please replot as bar graphs to show reduction in tumorsphere size at relevant doses vs EC50 curves.*

Response #6: Tumorspheres were chosen as the most biologically relevant *in vitro* model system for investigating drug responses, as brain tumor cultures grown in these stem cell-enriched conditions better recapitulate the genotypic and phenotypic changes observed in primary tumors than cells grown as adherent cultures in serum. Therefore, measurements of tumorsphere size in this preserved 3D state were considered to be a surrogate for effects on growth or proliferation.

To explain this further, we had stated in the original manuscript that our set-up was based on a modified EC50 as follows: *“To avoid inclusion of dead cells and aggregates, only true tumorspheres with diameters between 45 μ m and 360 μ m and circularities greater than 0.3 were scored. The dose-response curves were then used to generate a modified EC50 defined as the concentration that produces 50% tumorsphere size inhibition relative to the top and bottom sigmoidal constraints.”*

Indeed, we did not want to kill all the cells, as this would inhibit us from dissecting the effects on other cell properties such as tumorsphere number and migration. The screens were meant to identify a baseline concentration range for further testing of cell properties across multiple cell lines, and this includes additional follow-up studies on tumorsphere size using ImageJ/Fiji. While we believe the set-up and methods were valid, we appreciate the reviewer’s concern about determining EC50s. As such, we have converted these data to bar graphs and removed reference to EC50 as per the reviewer’s suggestion below in Rebuttal Figure 7 and in Figures 3e-f and Supplementary Figure 3d of the revised manuscript. Thank you to the reviewer for the suggestion.

Rebuttal Figure 7. Tumorsphere size analyses following treatment with various concentrations of Pacritinib (A), AZD1480 (B) or Vincristine (C). N=3 biological replicates for each treatment. Error bars: SEM. $p < 0.05$ *, $p < 0.01$ **, $p < 0.001$ ***.

7. Please include a western blot panel showing upregulation of pSTAT3 *in vitro* following selumetinib treatment *in vitro*.

Response #7: See response #3B above regarding comparison of models and corresponding Rebuttal Figure 5 demonstrating upregulation of pSTAT3 over 3 passages of Daoy tumorspheres treated with selumetinib *in vitro*.

8. For figure 4, it is also unclear why tumor size is the scoring parameter vs viability for the synergy studies. It is not convincing whether the reduction in tumour size is due to cell death or cell cycle arrest. The authors do go into a bit more granular assessment of viability and cell death in the supplementary data, however this really detracts from the main figure and is quite difficult to follow with multiple panels etc. It would be much easier to interpret the data if viability was the primary readout to begin with in the synergy studies.

Response #8: Please see response #6 above as justification for using tumorsphere size as opposed to viability as a readout.

9. All of the data in Figures 3 and 4 should be represented as normalized values to DMSO controls.

Response #9: While we appreciate the suggestion, respectfully, we feel that in order to be fully transparent, it is best to display raw, as opposed to normalized data for cellular assays throughout the manuscript.

10. The data in Figure 5 is misplaced as a main figure. The reviewer does not see any direct relevance of this analysis. It can be shifted as supplemental information.

Response #10: While we appreciate the suggestion, we feel that these data support the results from the cellular assays in Figure 4 and suggest a potential cytostatic mechanism in the combination therapy tumorsphere assays at the lower concentrations assessed. Moreover, as it is notoriously difficult to adapt primary SHH MB cells to long-term culture, we felt it was important to highlight the fact that genes/pathways associated with cell cycle and DNA replication/repair were the most significantly enriched downregulated gene sets following selumetinib or combination drug treatment in our cultured tumorsphere model. Strikingly, these pathways have also been shown to be strongly enriched specifically in high-risk SHH MB patient tumors exhibiting *TP53* mutations (Cavalli et al., 2017), and we believe this connection adds strength to the biological relevance of our model.

11. Total Duration of treatment is not clear for Figure 6a *in vivo* studies, please include in legend or directly on figure panel.

Response #11: We thank the reviewer for bringing this up and apologize for the lack of clarity on this matter. For all experiments, animals were treated twice daily, 5 days a week by oral gavage, with a 2-day drug holiday on weekends until they reached endpoint. For clarity, details described in the methods section have now been added to the Figure 6 legends on pages 33 of the revised manuscript.

12. Please include scale bars in Figure 7

Response #12: Scale bars have been added to all images and not just the first 2.

13. Number of animals used for the KM studies in figure 7 are very few for real KM studies

Response #13: Unfortunately, due to pandemic restrictions and multiple stops/re-starts in the *in vivo* experiments that were required over the course of 2020-2021, we could not perform all experiments with a large number of mice. This made completing the work for Figures 6, 7 and 8 quite difficult. However, we would like to point out that for the mouse weight data in Figure 6c, we tracked weights for 3 weeks following treatment initiation as a 1st sign of toxicity. As we were able to combine the weights from earlier time points from multiple studies, this allowed for an N=13-15 per treatment group, and we feel that this is substantial.

14. How is tumor size/burden determined? Authors need to provide details for methodology used for graphs in 7c and 7e

Response #14: Details have been added to the methods section on page 25 of the revised manuscript as follows:

“For tumor size calculations in both immunohistochemical (N=4 samples per treatment group) and MRI images (N=3 mice per treatment group), the ImageJ freehand tool was used to

determine area and volume respectively. STEM121 staining was used to demarcate the human tumor cell margins in immunohistochemical images. Tumor tissue, consisting of 1 or multiple lobes, in the cerebellum was visually identified on MRI images and used to delineate tumor size using the freehand tool. For each tumor sample, 18 serial sections (300 μm thickness) were obtained by MRI and volume calculated for each slice containing visible tumor tissue. The sum from all the slices containing tumor tissue was used to calculate an overall tumor volume.”

15. Significant toxicity has been previously reported with long term pacritinib treatment in mice vs humans, it is intriguing that the authors do not observe this in their model. They should elaborate on this a bit more in the discussion and also as per point 11, specify the total treatment duration.

Response #15: This is an excellent point. Indeed, previous studies (Jensen et al., Neuro-Oncology Advances, 2020; Jensen et al., PLoS One, 2017) have shown that combination therapies involving pacritinib have resulted in toxicity issues precluding long-term assessments of overall survival in mouse models. Managing toxicities while maintaining efficacy is a significant clinical challenge in the treatment of childhood cancer patients.

As pacritinib has been shown to be effective at lower concentrations, we focused on this drug in combination with selumetinib. Initial pilot studies in UI226 xenografts to determine the maximum tolerated dose revealed that 25 mg/kg, 50 mg/kg and 100 mg/kg of pacritinib were well-tolerated with no toxicities when administered via oral gavage 2X daily, 5 days a week until endpoint was reached. While not in the original version of the manuscript, the weight changes for these concentrations are now displayed in the new Figure 6a and complement the survival curve in Figure 6b of the revised manuscript. Weights are also shown below in Rebuttal Figure 8.

For the combination therapy studies, we chose the 50 mg/kg concentration of pacritinib as opposed to the 100 mg/kg utilized in the 2017 and 2020 Jensen studies. We believe that this lower dose, combined with the 5 day on, 2 day off “drug holiday” treatment schedule, helped to mitigate long-term toxicities in our model. However, similar to the Jensen studies, pacritinib alone did not improve survival in our animal models. Interestingly, the 2017 Jensen paper also demonstrated that pacritinib was rapidly metabolized by mouse, but not human liver microsomes. This rapid clearance, combined with lower overall JAK/STAT pathway activity in vehicle controls, may account for the lack of efficacy as a single agent in the xenografts. While we had briefly discussed toxicity in the original manuscript, we have now extended this section on page 16 of the revised manuscript to include the specific details outlined here. Thank you to the reviewer for the suggestions, as these statements improve the manuscript.

Rebuttal Figure 8. NOD SCID mouse weights over time following vehicle control (N=5), 25 mg/kg (N=4), 50 mg/kg (N=4) or 100 mg/kg (N=4) pacritinib treatment.

16. Figure 8E, what is the significance of ROIs between selutimenib or pacritinib treatment and the combinatorial treatment?

Response #16: The significance between selumetinib or pacritinib treatment and the combinatorial treatment for CD271 levels is provided below in Rebuttal Figure 9. This has also been updated in Figure 8f of the revised manuscript accordingly.

Rebuttal Figure 9. Boxplots depicting quantification of CD271 levels by mean fluorescent intensity across vehicle control, pacritinib, selumetinib and pacritinib + selumetinib treated samples. Bars represent minimum and maximum counts. $p < 0.05^*$, $p < 0.001^{***}$.

17. Why was pSTAT3 assessment not performed in the ROI proteomic studies, would be more relevant to the premise of this study.

Response #17: Yes, we completely agree that a JAK/STAT3 pathway panel would have been very informative for the spatial profiling studies. Unfortunately, only a certain number of rigorously tested and validated antibody panels were available to us through NanoString's technology testing program and were pertinent to the study. This includes a Neuro core panel, a MAPK panel, a PI3K panel and a cell death panel as indicated in the methods section of the manuscript. The Neuro core and cell death panels were helpful as a starting point to interrogate mechanism following combination drug treatment. A JAK/STAT3 panel was not an available option for this spatial profiling platform.

Reviewer #2:

In this paper, the authors identified the treatment by MEK inhibitor (selumetinib) decreased SHH MB growth while extending survival in mouse models. But the mice ultimately died from disease progression. Through RNA sequencing on selumetinib-treated orthotopic xenografts, the authors identified molecular pathways that compensate for MEK inhibition. They focused on JAK/STAT3 pathway due to its increased activation in selumetinib-treated tumors. The novel combination of the MEK and the JAK/STAT3 pathway inhibitors was reported to reduce tumor growth in two xenograft models and enhances survival. Multiplex spatial profiling of proteins in drug-treated xenografts reveals shifted molecular dependencies and compensatory changes following combination drug treatment. Taken together, this paper provided evidence showing MEK and JAK/STAT3 inhibition is a novel combinatory therapeutic strategy for SHH MB. This work is rigor and nicely illustrated. The idea of this work is initiated by the authors' own original findings. It is of interest to me and may be interesting to the wider field since the authors showed the potential of MEK and JAK/STAT3 combinatory therapy in the future, which may benefit the MB patients. Meanwhile, the digital spatial profiling reveals unique changes in proteins associated with MAPK signaling and cell death following combination therapy treatment, with which may inspire further studies and identify novel therapy target for all the subtypes of MB.

Response: We thank the reviewer for the insightful comments and appreciate that the reviewer found our work to be rigorous and nicely illustrated.

*Although this paper is of high quality, I still have concerns.
Major concerns:*

1. Some occasional grammar mistakes could be seen in the text. Careful language editing by native speakers is recommended.

Response #1: We thank the reviewer for the comment and have carefully edited the manuscript to correct these mistakes.

2. The authors established in vitro model of tumorspheres and PDX model for in vivo study. However, all the study models are SHH MB, the findings of this paper are not verified in the other subtypes of MB such as WNT, Group 3 and Group 4. While Group 3 and Group 4 MBs are well known to have much poorer prognostic and high recurrence rate. Thus, the title is recommended to be modified as "Combined MEK and JAK/STAT3 pathway inhibition effectively decreases SHH medulloblastoma tumor progression"

Response #2: We agree with the reviewer and have changed the title accordingly.

3. The toxicity and side effect of the dual MEK and JAK/STAT3 inhibition is not tested in this study. Discussion about this issue is necessary.

Response #3: We completely agree that additional discussion is necessary and thank the reviewer for the suggestion. In our experience with animal models, when drug concentrations are too high, we observe toxicities within the first 1-7 days of treatment. This translates to rapid weight decline and immediate termination of the experiment. We were very pleased that after

consultation with veterinary services and careful assessment of different concentrations for both the mono- and combination therapies, we did not observe any signs of toxicity. That being said, we acknowledge that necropsy was not performed on any of the organs following long-term treatment. This will be the subject of future studies.

As pacritinib has been shown to be effective at lower concentrations, we focused on this drug in combination with selumetinib. Initial pilot studies in UI226 xenografts to determine the maximum tolerated dose revealed that 25 mg/kg, 50 mg/kg and 100 mg/kg of pacritinib were well-tolerated with no toxicities when administered via oral gavage 2X daily, 5 days a week until endpoint was reached. While not in the original version of the manuscript, the weight changes for these concentrations are now displayed in the new Figure 6a and complement the survival curve in Figure 6b of the revised manuscript. Weights are also shown again in Rebuttal Figure 10 below.

Rebuttal Figure 10. NOD SCID mouse weights over time following vehicle control (N=5), 25 mg/kg (N=4), 50 mg/kg (N=4) or 100 mg/kg (N=4) pacritinib treatment.

For the combination therapy studies, we chose the 50 mg/kg concentration of pacritinib as opposed to the 100 mg/kg utilized in previous brain tumor studies (Jensen et al., *Neuro-Oncology Advances*, 2020; Jensen et al., *PLoS One*, 2017). We believe that this lower dose, combined with the 5 day on, 2 day off “drug holiday” treatment schedule, helped to mitigate long-term toxicities in our model. However, similar to the Jensen studies, pacritinib alone did not improve survival in our animal models. Interestingly, the 2017 Jensen paper also demonstrated that pacritinib was rapidly metabolized by mouse, but not human liver microsomes. This rapid clearance, combined with lower overall JAK/STAT pathway activity in vehicle controls, may account for the lack of efficacy as a single agent in the xenografts. While we had briefly discussed toxicity in the original manuscript, we have now extended this section on page 16 of the revised manuscript to include the specific details outlined here.

4. RNA sequencing for SHH MB in tumorspheres and digital spatial profiling for PDX models reveal significant molecular changes in SHH MB tumorspheres following dual MEK and JAK/STAT3 inhibition therapy. However, overlap of molecular changes between the in vivo and in vitro study is barely seen. Could you please explain this question in the discussion?

Response #4: This is an excellent point, and we agree that it should be clarified. Importantly, our Figure 1 and Figure 7 *in vivo* data are based on long-term (30-40+ days) treatment. Thus, it would not be appropriate to compare these results to tumorspheres in stem cell-enriched conditions over a very short 1-5 day period and expect all the same outcomes at a molecular level. As such, these experiments do not provide an adequate comparison to the *in vivo* data that assessed adaptive

responses long-term. The transcriptome changes in the tumorspheres corroborated our findings in Figure 4 that demonstrated significant decreases in tumorsphere size and migration. For example, changes in tumorsphere size are typically associated with growth factor responsiveness and proliferation (reviewed by Pastrana et al., Cell Stem Cell, 2011). Thus, it is not surprising that genes/pathways associated with cell cycle and DNA replication/repair were the most significantly enriched downregulated gene sets following selumetinib or combination drug treatment. Strikingly, these pathways have also been shown to be strongly enriched specifically in high-risk SHH MB patient tumors exhibiting *TP53* mutations (Cavalli et al. 2017). In support of our cell migration data in Figure 4, genes/pathways associated with epithelial to mesenchymal transition (EMT) were also significantly enriched among genes that were downregulated following selumetinib or combination drug treatment.

Regarding the DSP data, unfortunately, only a certain number of rigorously tested and validated antibody panels were available to us through NanoString's technology testing program and were pertinent to the study. This includes a Neuro core panel, a MAPK panel, a PI3K panel and a cell death panel as indicated in the methods section of the manuscript. The Neuro core and cell death panels were helpful as a starting point to interrogate mechanism following combination drug treatment. However, we acknowledge that this is a limitation of the study. We thank the reviewer for the comment and have now added the following statements to the discussion on pages 17-18 of the revised manuscript to add clarity on this matter.

“To our knowledge, this study is also one of the few to utilize multiplex spatial profiling to assess the impact of treatment on xenograft models, thus opening up new opportunities to comprehensively assess adaptive mechanisms and sustained drug responsiveness in situ. However, one limitation is the restricted number of relevant and validated antibody panels available for the spatial proteomics analyses. We were able to evaluate changes in proteins associated with neural differentiation, cell death, MAPK activity and PI3K activity following long-term combination drug treatment in vivo. While the data enabled an initial assessment of some molecular changes, a direct comparison to the RNA sequencing dataset from 5-day treated tumorspheres was difficult. Discrepancies between the short-term in vitro and long-term in vivo datasets are expected; however, future studies will expand the spatial profiling beyond proteomics to include comprehensive transcriptomic analyses to assess overlap.”

5. Regarding the MRI image in figure 7d. The authors declare that “combination drug treatment of RCMB18 SHH MB patient-derived xenografts (PDX) in a small cohort of NOD-SCID IL2Rg null (NSG) mice significantly reduced tumor growth, as only single well-encapsulated nodules were observed relative to vehicle controls.” But it seems that in the groups of pacritinib and pac+sel group, the sections are similar, but in the vehicle and sel group the section is higher, the same section of MRI is necessary. Additionally, in the pac group, the tumor is inerratic and encapsulated as well. Discussion is in need.

Response #5: The reviewer brings up an excellent point. Interestingly, this growth pattern was observed in the presence of pacritinib, with or without selumetinib. The tumors typically appeared as single well-encapsulated nodules as opposed to the vehicles and selumetinib-only samples which consisted of multiple infiltrating masses. This is an intriguing finding, and so we added this statement to the results section on page 11 of the revised manuscript. This also translated to different patterns on the MRIs with some tumors spanning 4-5 sections and others spanning 7-8 sections. Unfortunately, this also made it difficult to choose the best representative image for the figure. Thus, in the original submission, we chose the sections in which the tumors were the

largest for each sample. We realize that this can introduce bias, and so we have done the following:

1. We updated the images in Figure 7d with the same sections as much as possible.
2. Reviewer #3 (see below) suggested that we calculate tumor volume instead of tumor area for these images. Given the different tumor patterns observed following drug treatment, this would be the best way to eliminate any potential bias. We have 18 serial sections for each tumor. In each slide with visible tumor, we calculated the volume and added the totals from the slices together to generate total tumor volume for a given mouse. The results are displayed in Figure 7e of the revised manuscript as well as Rebuttal Figure 11 below for reference. RCMB18 is a bona fide *MYCN* amplified, *TP53* mutated SHH MB PDX model, and is thus classified as one of the “hard to treat,” drug resistant SHH subtypes. Therefore, the decrease in tumor growth following treatment with the combination therapy is promising.

The methods section has also been updated as follows on page 25 of the revised manuscript.

“For tumor size calculations in both immunohistochemical (N=4 samples per treatment group) and MRI images (N=3 mice per treatment group), the ImageJ freehand tool was used to determine area and volume respectively. STEM121 staining was used to demarcate the human tumor cell margins in immunohistochemical images. Tumor tissue, consisting of 1 or multiple lobes, in the cerebellum was visually identified on MRI images and used to delineate tumor size using the freehand tool. For each tumor sample, 18 serial sections (300 μm thickness) were obtained by MRI and volume calculated for each slice containing visible tumor tissue. The sum from all the slices containing tumor tissue was used to calculate an overall tumor volume.”

A

B

Rebuttal Figure 11. A. Representative MRI images from 3 independent RCMB18 tumors representing the vehicle control, pacritinib, selumetinib, and combination pacritinib + selumetinib treatment groups. Arrows denote individual or multiple tumor lobes in representative images from each treatment group. B. ImageJ quantification of RCMB18 tumor volume from compiled MRI images representing the vehicle control (N=3), pacritinib (N=3), selumetinib (N=3), and combination pacritinib + selumetinib (N=3) treatment groups. Error bars: SEM. $p < 0.05^*$.

Reviewer #3:

Zagozewski et al examine the effects of MEK and JAK/STAT inhibition on SHH-medulloblastoma tumor growth, both in vitro and in vivo. By examining RNA-seq data of MEKi-treated mice, the authors identify the JAK/STAT pathway as upregulated in treated animals compared with control animals. To further examine the role of JAK/STAT in MB growth, the authors first examine the effect of two JAK/STAT inhibitors on 2 human MB cell lines and 1 mouse MB cell line. After identifying pacritinib as effective in vitro, the authors perform combination treatment studies in vivo and show that lifespan of the animals is further extended, although the majority of animals eventually succumb to disease.

The manuscript flows nicely and in a logical manner, making the course of experiments easy to follow. It is certainly interesting how the JAK/STAT pathway is used by the tumor cells to compensate for MEK pathway inhibition, especially since this pathway isn't usually upregulated in SHH-MB patients. The low toxicity of the combination treatment also makes it a relevant therapeutic strategy and warrants future consideration.

Response: We thank the reviewer for the comments on the quality of our manuscript.

A few specific comments/concerns:

1. It is difficult to read/understand Fig 3A. Could you please make this figure larger and explain a bit more in the legends what the point of the figure is. Also for some of the legends, you do not state the statistical test used (For example Fig3G-L, but also others).

Response #1: We thank the reviewer for pointing this out and agree that it is hard to see. We have removed the heatmap and increased the size of the Hoechst 33342 stained tumorspheres. The purpose of this panel was to illustrate the drug screening set-up. We also agree that more details should be added and have now included the following in the legend on page 30-31 of the revised manuscript:

“Approximately 4,000 UI226 cells were seeded into each well of a 96-well ultra-low attachment plate. Cells were seeded in various drug concentrations or in the presence of vehicle (DMSO at 0.0025%) at 37°C for 5-days. Following incubation, tumorspheres were fixed and stained with a 4% formaldehyde + 300 ng/mL Hoechst 33342 solution at 4°C overnight. The following day, high-content imaging microscopy was performed.”

As suggested, we have now included the statistical test used in each panel for each figure legend.

2. I find it puzzling that AZD1480 seems to obliterate pSTAT3 at all concentrations tested (from the WB), also when compared with the pacritinib treated WB, where this effect is concentration dependent. However, the effect of AZD1480 treatment on tumorsphere size does not correspond to the levels of pSTAT3. Could it be that tumorsphere size is affected by something other than pSTAT inhibition (some off-target effect of pacritinib?) How can you reconcile these differences?

Response #2: We apologize for the confusion. The original pacritinib blots in Figure 3 and AZD1480 blots in Supplementary Figure 3 were performed at different time points using different concentration ranges. The very high concentrations used for the AZD1480 blots in our original submission explain the negligible levels of pSTAT3. This was a mistake on our part, as the blots should be consistent. We have now performed new experiments for AZD1480 at the same 3 hour

time point with the same concentration range as pacritinib. These new data are displayed in Supplementary Figure 3 and in Rebuttal Figure 12 below for clarity.

3. Along those same lines – are you sure that the *ptch*;*p53* mouse cells have pSTAT at basal levels? Perhaps it would be useful to show a similar WB for those cells, assuming you have material/antibodies.

Response #3: We apologize for the omission. Although we had limited material from these primary mouse cultures, at the time of submission, we were struggling with antibody compatibility and could not include the data. We have since optimized the western blot conditions for the *Ptch*^{+/-};*p53*^{+/-} mouse primary Shh MB tumor cells. Results from these new experiments are displayed in Figure 3d and Supplementary Figure 3c of the revised manuscript as well as Rebuttal Figure 12 below for clarity. Indeed, the *Ptch*^{+/-};*p53*^{+/-} mouse primary Shh MB tumor cells exhibit pSTAT3 at basal levels. It is well known in the field that it is extremely difficult to adapt primary SHH MB cells to long-term culture. Very few models exist for *in vitro* studies. These data strengthen our manuscript by showing the effects of multiple inhibitors across 3 different tumorsphere models including 2 cell lines and 1 low passage primary culture.

Rebuttal Figure 12. Western blots depicting a decrease in pSTAT3 following treatment with pacritinib or AZD1480. A-C. Western blot for pSTAT3 (Tyr705) activation, total STAT3 and GAPDH following treatment of UI226 (A) Daoy (B) or *Ptch*^{+/-};*p53*^{+/-} (C) Shh MB tumorspheres with pacritinib for 3 hours. D-F. Western blot for pSTAT3 (Tyr705) activation, total STAT3 and GAPDH following treatment of UI226 (D) Daoy (E) or *Ptch*^{+/-};*p53*^{+/-} (F) Shh MB tumorspheres with AZD1480 for 3 hours.

4. In figure 7, I'm surprised you still have neurological signs of tumor formation with such small tumors (pac+sel treated). Are these end point tumors? If not, when were these mice sacrificed in relation to treatment/tumor growth? Does this mean the tumor grows back after initially shrinking? For Figure 7, I also think estimating tumor volume (not area) would be more helpful, since area relies on where you cut in FFPE sections. Can you use the FFPE blocks to estimate tumor volume? Same for the RCMB18 – can you estimate tumor volume on the MRI instead of area? For these images, it would also be helpful for those not used to looking at MRIs to mark on the figures where the tumor is located.

Response #4: We thank the reviewer for the very valuable suggestions. For the growth assessments in Figure 7, we had taken a subset of the UI226 xenografts, both vehicle and drug-treated, prior to endpoint, when the controls started showing signs of tumor progression. Of note, these same samples were used for the digital spatial profiling studies in Figure 8. Thus, we believe that treatment delays or slows tumor growth and as such, the animals take longer to reach endpoint. Similarly, the RCMB18 were taken prior to endpoint to assess growth at the same time point across samples. We apologize for the confusion and thank the reviewer for bringing this up. We have now updated the text to clarify as follows:

“To compare differences in tumor growth across treatment groups, a subset of UI226 vehicle and treated xenografts was also extracted prior to endpoint, all at day 41, and samples were stained for the human specific marker STEM121 (Fig. 7a-b). While a small decrease in tumor size was observed for single agent therapies, combination drug treatment resulted in a substantial and significant reduction in tumor growth (Fig. 7a-c).”

Regarding area vs. volume, the reviewer brings up an excellent point. Unfortunately, the blocks from the UI226 samples are no longer available, so we are unable to have more sections cut to assess volume in place of area for Figure 7c. However, we are able to assess volume for our MRI images. We have 18 serial sections obtained from MRI per tumor. For each image with visible tumor in a given mouse, we calculated the volume and added the totals from the slices together to generate total tumor volume.

The methods section has been updated as follows on page 25 of the revised manuscript.

“For tumor size calculations in both immunohistochemical (N=4 samples per treatment group) and MRI images (N=3 mice per treatment group), the ImageJ freehand tool was used to determine area and volume respectively. STEM121 staining was used to demarcate the human tumor cell margins in immunohistochemical images. Tumor tissue, consisting of 1 or multiple lobes, in the cerebellum was visually identified on MRI images and used to delineate tumor size using the freehand tool. For each tumor sample, 18 serial sections (300 μm thickness) were obtained by MRI and volume calculated for each slice containing visible tumor tissue. The sum from all the slices containing tumor tissue was used to calculate an overall tumor volume.”

The new data have been added to Figure 7e of the revised manuscript and are also presented below in Rebuttal Figure 13 for reference. As suggested, we have also added arrows to demarcate the tumors in representative images from each treatment group.

Rebuttal Figure 13. ImageJ quantification of RCMB18 tumor volume from compiled MRI images representing the vehicle control (N=3), pacritinib (N=3), selumetinib (N=3), and combination pacritinib + selumetinib (N=3) treatment groups. Error bars: SEM. $p < 0.05^*$.

REVIEWERS' COMMENTS:

Reviewer #1 (Remarks to the Author):

The authors have carefully revised the manuscript to address the majority of my concerns. A couple of remaining points if addressed will fully address a couple of lingering points.

1. In response to 3B, please add Rebuttal Figures 4 and 5 to the supplemental Figures.
2. In response to question 9, if the authors present raw data they should include data on untreated cells in all of the relevant graphs and images as well to rule out any DMSO effects which are not obvious otherwise

Reviewer #3 (Remarks to the Author):

All of my concerns were addressed. Only minor point is that Figure 1G is no longer representative (it looks nearly 0% vs 100%, when the quantification suggests 10% vs 25%). It might be better to choose a better representative image instead.

Response to reviewers:

Reviewer #1:

1. In response to 3B, please add Rebuttal Figures 4 and 5 to the supplemental Figures.

Rebuttal Figures 4 and 5 have now been included in the manuscript as Supplementary Fig. 1b, and Supplementary Fig. 1c, d, respectively.

2. In response to question 9, if the authors present raw data they should include data on untreated cells in all of the relevant graphs and images as well to rule out any DMSO effects which are not obvious otherwise

Thank you for your comment. Respectively, we believe that repetition of existing experiments to include data on untreated cells would not yield meaningful results that would change the overall findings of our manuscript, nor do we anticipate these experiments could be completed in a reasonable time frame for publication. At this point, we feel that it is important to leave the figures "as is" in raw data format. We don't believe that normalizing the data, as originally requested, would answer the question regarding DMSO effects, especially given that all the data, including westerns blots and RNA sequencing in tumorspheres etc. are displayed relative to DMSO. Therefore, we respectfully request leaving the figures in their current format, which already include all the individual data points for transparency.

Reviewer #3:

All of my concerns were addressed. Only minor point is that Figure 1G is no longer representative (it looks nearly 0% vs 100%, when the quantification suggests 10% vs 25%). It might be better to choose a better representative image instead.

Thank you to the reviewer for your comment. We have now updated Figure 1g to include more representative images better depicting the 2.5 fold increase quantified in Figure 1h.